# Flat Seeking Bayesian Neural Networks

**Van-Anh Nguyen**[1]   **Tung-Long Vuong**[1,2]   **Hoang Phan**[2,3]   **Thanh-Toan Do**[1]

**Dinh Phung** [1,2]   **Trung Le** [1]

[1]Department of Data Science and AI, Monash University, Australia

[2]VinAI, Vietnam

[3]New York University, United States

{van-anh.nguyen, tung-long.vuong, toan.do, dinh.phung, trunglm}@monash.edu

hvp2011@nyu.edu

## Abstract

Bayesian Neural Networks (BNNs) provide a probabilistic interpretation for deep learning models by imposing a prior distribution over model parameters and inferring a posterior distribution based on observed data. The model sampled from the posterior distribution can be used for providing ensemble predictions and quantifying prediction uncertainty. It is well-known that deep learning models with lower sharpness have better generalization ability. However, existing posterior inferences are not aware of sharpness/flatness in terms of formulation, possibly leading to high sharpness for the models sampled from them. In this paper, we develop theories, the Bayesian setting, and the variational inference approach for the sharpness-aware posterior. Specifically, the models sampled from our sharpness-aware posterior, and the optimal approximate posterior estimating this sharpness-aware posterior, have better flatness, hence possibly possessing higher generalization ability. We conduct experiments by leveraging the sharpness-aware posterior with state-of-the-art Bayesian Neural Networks, showing that the flat-seeking counterparts outperform their baselines in all metrics of interest.

## 1   Introduction

Bayesian Neural Networks (BNNs) provide a way to interpret deep learning models probabilistically. This is done by setting a prior distribution over model parameters and then inferring a posterior distribution over model parameters based on observed data. This allows us to not only make predictions, but also quantify prediction uncertainty, which is useful for many real-world applications. To sample deep learning models from complex and complicated posterior distributions, advanced particle-sampling approaches such as Hamiltonian Monte Carlo (HMC) [41], Stochastic Gradient HMC (SGHMC) [10], Stochastic Gradient Langevin dynamics (SGLD) [58], and Stein Variational Gradient Descent (SVGD) [36] are often used. However, these methods can be computationally expensive, particularly when many models need to be sampled for better ensembles.

To alleviate this computational burden and enable the sampling of multiple deep learning models from posterior distributions, variational inference approaches employ approximate posteriors to estimate the true posterior. These methods utilize approximate posteriors that belong to sufficiently rich families, which are both economical and convenient to sample from. However, the pioneering works in variational inference, such as [21, 5, 33], assume approximate posteriors to be fully factorized distributions, also known as mean-field variational inference. This approach fails to account for the strong statistical dependencies among random weights of neural networks, limiting its ability to capture the complex structure of the true posterior and estimate the true model uncertainty. To overcome this issue, latter works have attempted to provide posterior approximations with richer

37th Conference on Neural Information Processing Systems (NeurIPS 2023).

expressiveness [61, 52, 53, 54, 20, 45, 55, 30, 48]. These approaches aim to improve the accuracy of the posterior approximation and enable more effective uncertainty quantification.

In the context of standard deep network training, it has been observed that flat minimizers can enhance the generalization capability of models. This is achieved by enabling them to locate wider local minima that are more robust to shifts between train and test sets. Several studies, including [27, 47, 15], have shown evidence to support this principle. However, the posteriors used in existing Bayesian neural networks (BNNs) do not account for the sharpness/flatness of the models derived from them in terms of model formulation. As a result, the sampled models can be located in regions of high sharpness and low flatness, leading to poor generalization ability. Moreover, in variational inference methods, using approximate posteriors to estimate these non-sharpness-aware posteriors can result in sampled models from the corresponding optimal approximate posterior lacking awareness of sharpness/flatness, hence causing them to suffer from poor generalization ability.

In this paper, our objective is to propose a sharpness-aware posterior for learning BNNs, which samples models with high flatness for better generalization ability. To achieve this, we devise both a Bayesian setting and a variational inference approach for the proposed posterior. By estimating the optimal approximate posteriors, we can generate flatter models that improve the generalization ability. Our approach is as follows: In Theorem 3.1, we show that the standard posterior is the optimal solution to an optimization problem that balances the empirical loss induced by models sampled from an approximate posterior for fitting a training set with a Kullback-Leibler (KL) divergence, which encourages a simple approximate posterior. Based on this insight, we replace the empirical loss induced by the approximate posterior with the general loss over the entire data-label distribution in Theorem 3.2 to improve the generalization ability. Inspired by sharpness-aware minimization [16], we develop an upper-bound of the general loss in Theorem 3.2, leading us to formulate the sharpness-aware posterior in Theorem 3.3. Finally, we devise the Bayesian setting and variational approach for the sharpness-aware posterior. Overall, our contributions in this paper can be summarized as follows:

- We propose and develop theories, the Bayesian setting, and the variational inference approach for the sharpness-aware posterior. This posterior enables us to sample a set of flat models that improve the model generalization ability. We note that SAM [16] only considers the sharpness for a single model, while ours is the first work studying the concept and theory of the sharpness for a distribution $\mathbb{Q}$ over models. Additionally, the proof of Theorem 3.2 is very challenging, elegant, and complicated because of the infinite number of models in the support of $\mathbb{Q}$.

- We conduct extensive experiments by leveraging our sharpness-aware posterior with the state-of-the-art and well-known BNNs, including *BNNs with an approximate Gaussian distribution [33]*, *BNNs with stochastic gradient Langevin dynamics (SGLD) [58]*, *MC-Dropout [18]*, *Bayesian deep ensemble [35]*, and *SWAG [39]* to demonstrate that the flat-seeking counterparts consistently outperform the corresponding approaches in all metrics of interest, including the ensemble accuracy, expected calibration error (ECE), and negative log-likelihood (NLL).

## 2  Related Work

### 2.1  Bayesian Neural Networks

**Markov chain Monte Carlo (MCMC):** This approach allows us to sample multiple models from the posterior distribution and was well-known for inference with neural networks through the Hamiltonian Monte Carlo (HMC) [41]. However, HMC requires the estimation of full gradients, which is computationally expensive for neural networks. To make the HMC framework practical, Stochastic Gradient HMC (SGHMC) [10] enables stochastic gradients to be used in Bayesian inference, crucial for both scalability and exploring a space of solutions. Alternatively, stochastic gradient Langevin dynamics (SGLD) [58] employs first-order Langevin dynamics in the stochastic gradient setting. Additionally, Stein Variational Gradient Descent (SVGD) [36] maintains a set of particles to gradually approach a posterior distribution. Theoretically, all SGHMC, SGLD, and SVGD asymptotically sample from the posterior in the limit of infinitely small step sizes.

**Variational Inference**: This approach uses an approximate posterior distribution in a family to estimate the true posterior distribution by maximizing a variational lower bound. [21] suggests fitting

a Gaussian variational posterior approximation over the weights of neural networks, which was generalized in [32, 33, 5], using the reparameterization trick for training deep latent variable models. To provide posterior approximations with richer expressiveness, many extensive studies have been proposed. Notably, [38] treats the weight matrix as a whole via a matrix variate Gaussian [22] and approximates the posterior based on this parameterization. Several later works have inspected this distribution to examine different structured representations for the variational Gaussian posterior, such as Kronecker-factored [59, 52, 53], k-tied distribution [54], non-centered or rank-1 parameterization [20, 14]. Another recipe to represent the true covariance matrix of Gaussian posterior is through the low-rank approximation [45, 55, 30, 39].

**Dropout Variational Inference:** This approach utilizes dropout to characterize approximate posteriors. Typically, [18] and [33] use this principle to propose Bayesian Dropout inference methods such as MC Dropout and Variational Dropout. Concrete dropout [19] extends this idea to optimize the dropout probabilities. Variational Structured Dropout [43] employs Householder transformation to learn a structured representation for multiplicative Gaussian noise in the Variational Dropout method.

## 2.2 Flat Minima

Flat minimizers have been found to improve the generalization ability of neural networks. This is because they enable models to find wider local minima, which makes them more robust against shifts between train and test sets [27, 47, 15, 44]. The relationship between generalization ability and the width of minima has been investigated theoretically and empirically in many studies, notably [23, 42, 12, 17]. Moreover, various methods seeking flat minima have been proposed in [46, 9, 29, 25, 16, 44]. Typically, [29, 26, 57] investigate the impacts of different training factors such as batch size, learning rate, covariance of gradient, and dropout on the flatness of found minima. Additionally, several approaches pursue wide local minima by adding regularization terms to the loss function [46, 61, 60, 9]. Examples of such regularization terms include softmax output's low entropy penalty [46] and distillation losses [61, 60].

SAM, a method that aims to minimize the worst-case loss around the current model by seeking flat regions, has recently gained attention due to its scalability and effectiveness compared to previous methods [16, 56]. SAM has been widely applied in various domains and tasks, such as meta-learning bi-level optimization [1], federated learning [51], multi-task learning [50], where it achieved tighter convergence rates and proposed generalization bounds. SAM has also demonstrated its generalization ability in vision models [11], language models [3], domain generalization [8], and multi-task learning [50]. Some researchers have attempted to improve SAM by exploiting its geometry [34, 31], additionally minimizing the surrogate gap [62], and speeding up its training time [13, 37]. Regarding the behavior of SAM, [28] empirically studied the difference in sharpness obtained by SAM [16] and SWA [24], [40] showed that SAM is an optimal Bayes relaxation of the standard Bayesian inference with a normal posterior, while [44] proved that distribution robustness [4, 49] is a probabilistic extension of SAM.

## 3 Proposed Framework

In what follows, we present the technicality of our proposed sharpness-aware posterior. Particularly, Section 3.1 introduces the problem setting and motivation for our sharpness-aware posterior. Section 3.2 is dedicated to our theory development, while Section 3.3 is used to describe the Bayesian setting and variational inference approach for our sharpness-aware posterior.

### 3.1 Problem Setting and Motivation

We aim to develop Sharpness-Aware Bayesian Neural Networks (SA-BNN). Consider a family of neural networks $f_\theta(x)$ with $\theta \in \Theta$ and a training set $\mathcal{S} = \{(x_1, y_1), ..., (x_n, y_n)\}$ where $(x_i, y_i) \sim \mathcal{D}$. We wish to learn a posterior distribution $\mathbb{Q}_\mathcal{S}^{SA}$ with the density function $q^{SA}(\theta|\mathcal{S})$ such that any model $\theta \sim \mathbb{Q}_\mathcal{S}^{SA}$ is aware of the sharpness when predicting over the training set $\mathcal{S}$.

We depart with the standard posterior

$$q(\theta \mid \mathcal{S}) \propto \prod_{i=1}^{n} p(y_i \mid x_i, \mathcal{S}, \theta) p(\theta),$$

where the prior distribution $\mathbb{P}$ has the density function $p(\theta)$ and the likelihood has the form

$$p(y \mid x, \mathcal{S}, \theta) \propto \exp\left\{-\frac{\lambda}{|\mathcal{S}|}\ell\left(f_\theta(x), y\right)\right\} = \exp\left\{-\frac{\lambda}{n}\ell\left(f_\theta(x), y\right)\right\}$$

with the loss function $\ell$. The standard posterior $\mathbb{Q}_\mathcal{S}$ has the density function defined as

$$q(\theta \mid \mathcal{S}) \propto \exp\left\{-\frac{\lambda}{n}\sum_{i=1}^{n}\ell\left(f_\theta\left(x_i\right), y_i\right)\right\}p(\theta), \tag{1}$$

where $\lambda \geq 0$ is a regularization parameter.

We define the general and empirical losses as follows:

$$\mathcal{L}_\mathcal{D}\left(\theta\right) = \mathbb{E}_{(x,y)\sim\mathcal{D}}\left[\ell\left(f_\theta\left(x\right), y\right)\right].$$

$$\mathcal{L}_\mathcal{S}\left(\theta\right) = \mathbb{E}_{(x,y)\sim\mathcal{S}}\left[\ell\left(f_\theta\left(x\right), y\right)\right] = \frac{1}{n}\sum_{i=1}^{n}\ell\left(f_\theta\left(x_i\right), y_i\right).$$

Basically, the general loss is defined as the expected loss over the entire data-label distribution $\mathcal{D}$, while the empirical loss is defined as the empirical loss over a specific training set $\mathcal{S}$.

The standard posterior in Eq. (1) can be rewritten as

$$q(\theta \mid \mathcal{S}) \propto \exp\left\{-\lambda\mathcal{L}_\mathcal{S}\left(\theta\right)\right\}p(\theta). \tag{2}$$

Given a distribution $\mathbb{Q}$ with the density function $q\left(\theta\right)$ over the model parameters $\theta \in \Theta$, we define the empirical and general losses over this model distribution $\mathbb{Q}$ as

$$\mathcal{L}_\mathcal{S}\left(\mathbb{Q}\right) = \int_\Theta \mathcal{L}_\mathcal{S}\left(\theta\right)d\mathbb{Q}\left(\theta\right) = \int_\Theta \mathcal{L}_\mathcal{S}\left(\theta\right)q\left(\theta\right)d\theta.$$

$$\mathcal{L}_\mathcal{D}\left(\mathbb{Q}\right) = \int_\Theta \mathcal{L}_\mathcal{D}\left(\theta\right)d\mathbb{Q}\left(\theta\right) = \int_\Theta \mathcal{L}_\mathcal{D}\left(\theta\right)q\left(\theta\right)d\theta.$$

Specifically, the general loss over the model distribution $\mathbb{Q}$ is defined as the expectation of the general losses incurred by the models sampled from this distribution, while the empirical loss over the model distribution $\mathbb{Q}$ is defined as the expectation of the empirical losses incurred by the models sampled from this distribution.

## 3.2   Our Theory Development

We now present the theory development for the sharpness-aware posterior whose proofs can be found in the supplementary material. Inspired by the Gibbs form of the standard posterior $\mathbb{Q}_\mathcal{S}$ in Eq. (2), we establish the following theorem to connect the standard posterior $\mathbb{Q}_\mathcal{S}$ with the density $q(\theta \mid \mathcal{S})$ and the empirical loss $\mathcal{L}_\mathcal{S}\left(\mathbb{Q}\right)$ [7, 2].

**Theorem 3.1.** *Consider the following optimization problem*

$$\min_{\mathbb{Q}<<\mathbb{P}}\left\{\lambda\mathcal{L}_S\left(\mathbb{Q}\right) + KL\left(\mathbb{Q}, \mathbb{P}\right)\right\}, \tag{3}$$

*where we search over $\mathbb{Q}$ absolutely continuous w.r.t. $\mathbb{P}$ and $KL\left(\cdot, \cdot\right)$ is the Kullback-Leibler divergence. This optimization has a closed-form optimal solution $\mathbb{Q}^*$ with the density*

$$q^*\left(\theta\right) \propto \exp\left\{-\lambda\mathcal{L}_S\left(\theta\right)\right\}p(\theta),$$

*which is exactly the standard posterior $\mathbb{Q}_\mathcal{S}$ with the density $q(\theta \mid \mathcal{S})$.*

Theorem 3.1 reveals that we need to find the posterior $\mathbb{Q}_\mathcal{S}$ balancing between optimizing its empirical loss $\mathcal{L}_S\left(\mathbb{Q}\right)$ and simplicity via $KL\left(\mathbb{Q}, \mathbb{P}\right)$. However, minimizing the empirical loss $\mathcal{L}_S\left(\mathbb{Q}\right)$ only ensures the correct predictions for the training examples in $\mathcal{S}$, hence possibly encountering overfitting. Therefore, it is desirable to replace the empirical loss by the general loss to combat overfitting.

To mitigate overfitting, in (3), we replace the empirical loss by the general loss and solve the following optimization problem (OP):

$$\min_{\mathbb{Q}<<\mathbb{P}}\left\{\lambda\mathcal{L}_\mathcal{D}\left(\mathbb{Q}\right) + KL\left(\mathbb{Q}, \mathbb{P}\right)\right\}. \tag{4}$$

Notably, solving the optimization problem (OP) in (4) is generally intractable. To make it tractable, we find its upper-bound which is relevant to the sharpness of a distribution $\mathbb{Q}$ over models as shown in the following theorem.

**Theorem 3.2.** *Assume that $\Theta$ is a compact set. Under some mild conditions, given any $\delta \in [0; 1]$, with the probability at least $1 - \delta$ over the choice of $\mathcal{S} \sim \mathcal{D}^n$, for any distribution $\mathbb{Q}$, we have*

$$\mathcal{L}_{\mathcal{D}}\left(\mathbb{Q}\right) \leq \mathbb{E}_{\theta \sim \mathbb{Q}}\left[\max_{\theta' : \|\theta' - \theta\| \leq \rho} \mathcal{L}_{\mathcal{S}}\left(\theta'\right)\right] + f\left(\max_{\theta \in \Theta} \|\theta\|^2, n\right),$$

*where $f$ is a non-decreasing function w.r.t. the first variable and approaches $0$ when the training size $n$ approaches $\infty$.*

We note that the proof of Theorem 3.2 is not a trivial extension of sharpness-aware minimization because we need to tackle the general and empirical losses over a distribution $\mathbb{Q}$. To make explicit our sharpness over a distribution $\mathbb{Q}$ on models, we rewrite the upper-bound of the inequality as

$$\mathbb{E}_{\theta \sim \mathbb{Q}}\left[\max_{\theta' : \|\theta' - \theta\| \leq \rho} \mathcal{L}_{\mathcal{S}}\left(\theta'\right) - \mathcal{L}_{\mathcal{S}}\left(\theta\right)\right] + \mathcal{L}_{\mathcal{S}}\left(\mathbb{Q}\right) + f\left(\max_{\theta \in \Theta} \|\theta\|^2, n\right),$$

where the first term $\mathbb{E}_{\theta \sim \mathbb{Q}}\left[\max_{\theta' : \|\theta' - \theta\| \leq \rho} \mathcal{L}_{\mathcal{S}}\left(\theta'\right) - \mathcal{L}_{\mathcal{S}}\left(\theta\right)\right]$ can be regarded as *the sharpness over the distribution $\mathbb{Q}$ on the model space* and the last term $f\left(\max_{\theta \in \Theta} \|\theta\|^2, n\right)$ is a constant.

Moreover, inspired by Theorem 3.2, we propose solving the following OP which forms an upper-bound of the desirable OP in (4)

$$\min_{\mathbb{Q} << \mathbb{P}} \left\{ \lambda \mathbb{E}_{\theta \sim \mathbb{Q}}\left[\max_{\theta' : \|\theta' - \theta\| \leq \rho} \mathcal{L}_{\mathcal{S}}\left(\theta'\right)\right] + KL\left(\mathbb{Q}, \mathbb{P}\right) \right\}. \tag{5}$$

The following theorem characterizes the optimal solution of the OP in (5).

**Theorem 3.3.** *The optimal solution the OP in (5) is the sharpness-aware posterior distribution $\mathbb{Q}_S^{SA}$ with the density function $q^{SA}(\theta|\mathcal{S})$:*

$$q^{SA}(\theta|\mathcal{S}) \propto \exp\left\{-\lambda \max_{\theta' : \|\theta' - \theta\| \leq \rho} \mathcal{L}_{\mathcal{S}}\left(\theta'\right)\right\} p\left(\theta\right) = \exp\left\{-\lambda \mathcal{L}_{\mathcal{S}}\left(s\left(\theta\right)\right)\right\} p\left(\theta\right),$$

*where we have defined $s\left(\theta\right) = \underset{\theta' : \|\theta' - \theta\| \leq \rho}{argmax} \mathcal{L}_{\mathcal{S}}\left(\theta'\right)$.*

Theorem 3.3 describes the close form of the sharpness-aware posterior distribution $\mathbb{Q}_S^{SA}$ with the density function $q^{SA}(\theta|\mathcal{S})$. Based on this characterization, in what follows, we introduce the SA Bayesian setting that sheds lights on its variational approach.

### 3.3 Sharpness-Aware Bayesian Setting and Its Variational Approach

**Bayesian Setting:** To promote the Bayesian setting for sharpness-aware posterior distribution $\mathbb{Q}_S^{SA}$, we examine the sharpness-aware likelihood

$$p^{SA}\left(y \mid x, \mathcal{S}, \theta\right) \propto \exp\left\{-\frac{\lambda}{|\mathcal{S}|} \ell\left(f_{s(\theta)}(x), y\right)\right\} = \exp\left\{-\frac{\lambda}{n} \ell\left(f_{s(\theta)}(x), y\right)\right\},$$

where $s\left(\theta\right) = \underset{\theta' : \|\theta' - \theta\| \leq \rho}{argmax} \mathcal{L}_{\mathcal{S}}\left(\theta'\right)$.

With this predefined sharpness-aware likelihood, we can recover the sharpness-aware posterior distribution $\mathbb{Q}_S^{SA}$ with the density function $q^{SA}(\theta|\mathcal{S})$:

$$q^{SA}(\theta|\mathcal{S}) \propto \prod_{i=1}^{n} p^{SA}\left(y_i \mid x_i, \mathcal{S}, \theta\right) p\left(\theta\right).$$

**Variational inference for the sharpness-aware posterior distribution:** We now develop the variational inference for the sharpness-aware posterior distribution. Let denote $X = [x_1, ..., x_n]$ and

$Y = [y_1, ..., y_n]$. Considering an approximate posterior family $\{q_\phi(\theta) : \phi \in \Phi\}$, we have

$$\log p^{SA}(Y \mid X, \mathcal{S}) = \int_\Theta q_\phi(\theta) \log p^{SA}(Y \mid X, \mathcal{S}) \, d\theta$$

$$= \int_\Theta q_\phi(\theta) \log \frac{p^{SA}(Y \mid \theta, X, \mathcal{S}) \, p(\theta)}{q_\phi(\theta)} \frac{q_\phi(\theta)}{q^{SA}(\theta|\mathcal{S})} \, d\theta$$

$$= \mathbb{E}_{q_\phi(\theta)} \left[ \sum_{i=1}^n \log p^{SA}(y_i \mid x_i, \mathcal{S}, \theta) \right] - KL(q_\phi, p) + KL(q_\phi, q^{SA}).$$

It is obvious that we need to maximize the following lower bound for maximally reducing the gap $KL(q_\phi, q^{SA})$:

$$\max_{q_\phi} \left\{ \mathbb{E}_{q_\phi(\theta)} \left[ \sum_{i=1}^n \log p^{SA}(y_i \mid x_i, \mathcal{S}, \theta) \right] - KL(q_\phi, p) \right\},$$

which can be equivalently rewritten as

$$\min_{q_\phi} \left\{ \lambda \mathbb{E}_{q_\phi(\theta)} \left[ \mathcal{L}_\mathcal{S}(s(\theta)) \right] + KL(q_\phi, p) \right\} \text{ or}$$

$$\min_{q_\phi} \left\{ \lambda \mathbb{E}_{q_\phi(\theta)} \left[ \max_{\theta': \|\theta' - \theta\| \leq \rho} \mathcal{L}_\mathcal{S}(\theta') \right] + KL(q_\phi, p) \right\}. \tag{6}$$

**Derivation for Variational Approach with A Gaussian Approximate Posterior:** Inspired by the geometry-based SAM approaches [34, 31], we incorporate the geometry to the SA variational approach via the distance to define the ball for the sharpness as $\|\theta' - \theta\|_{\text{diag}(T_\theta)} = \sqrt{(\theta' - \theta)^T \text{diag}(T_\theta)^{-1} (\theta' - \theta)}$ as

$$\min_{q_\phi} \left\{ \lambda \mathbb{E}_{q_\phi(\theta)} \left[ \max_{\theta': \|\theta' - \theta\|_{\text{diag}(T_\theta)} \leq \rho} \mathcal{L}_\mathcal{S}(\theta') \right] + KL(q_\phi, p) \right\}.$$

To further clarify, we consider our SA posterior distribution to Bayesian NNs, wherein we impose the Gaussian distributions to its weight matrices $W_i \sim \mathcal{N}(\mu_i, \sigma_i^2 \mathbb{I}), i = 1, \ldots, L$[1]. The parameter $\phi$ consists of $\mu_i, \sigma_i, i = 1, \ldots, L$. For $\theta = W_{1:L} \sim q_\phi$, using the reparameterization trick $W_i = \mu_i + \text{diag}(\sigma_i)\epsilon_i, \epsilon_i \sim \mathcal{N}(0, \mathbb{I})$ and by searching $\theta' = W'_{1:L}$ with $W'_i = \mu'_i + \text{diag}(\sigma_i)\epsilon_i, \epsilon_i \sim \mathcal{N}(0, \mathbb{I})$, the constraint $\|\theta - \theta'\|_{\text{diag}(T_\theta)} = \|\mu - \mu'\|_{\text{diag}(T_\theta)}$ with $\mu = \mu_{1:L}$ and $\mu' = \mu'_{1:L}$. Thus, the OP in (6) reads

$$\min_{\mu, \sigma} \left\{ \lambda \mathbb{E}_\epsilon \left[ \max_{\|\mu' - \mu\|_{\text{diag}(T_{\mu, \sigma})} \leq \rho} \mathcal{L}_\mathcal{S} \left( \left[ \mu'_i + \text{diag}(\sigma_i)\epsilon_i \right]_{i=1}^L \right) \right] \right\}, \tag{7}$$

where $\sigma = \sigma_{1:L}$, $\epsilon = \epsilon_{1:L}$, and we define $\text{diag}(T_\theta) = \text{diag}(T_{\mu, \sigma})$ in the distance of the geometry.

To solve the OP in (7), we sample $\epsilon = \epsilon_{1:L}$ from the standard Gaussian distributions, employ an one-step gradient ascent to find $\mu'$, and use the gradient at $\mu'$ to update $\mu$. Specifically, we find $\mu'$ [6] (Chapter 9) as

$$\mu' = \mu + \rho \frac{\text{diag}(T_{\mu, \sigma}) \nabla_\mu \mathcal{L}_\mathcal{S} \left( [\mu_i + \text{diag}(\sigma_i)\epsilon_i]_{i=1}^L \right)}{\left\| \text{diag}(T_{\mu, \sigma}) \nabla_\mu \mathcal{L}_\mathcal{S} \left( [\mu_i + \text{diag}(\sigma_i)\epsilon_i]_{i=1}^L \right) \right\|}.$$

The diagnose of $\text{diag}(T_{\mu, \sigma})$ specifies the importance level of the model weights, i.e., the weight with a higher importance level is encouraged to have a higher sharpness via a smaller absolute partial derivative of the loss w.r.t. this weight. We consider $\text{diag}(T_{\mu, \sigma}) = \mathbb{I}$ (i.e., *the standard SA BNN*) and $\text{diag}(T_{\mu, \sigma}) = \text{diag}\left(\frac{|\mu|}{\sigma}\right)$ (i.e., *the geometry SA BNN*). Here we note that $\vdots$ represents the element-wise division.

---

[1] We absorb the biases to the weight matrices.

Table 1: Classification score on CIFAR-100 dataset. Each experiment is repeated three times with different random seeds and reports the mean and standard deviation.

| | PreResNet-164 | | | WideResNet28x10 | | |
|---|---|---|---|---|---|---|
| Method | ACC ↑ | NLL ↓ | ECE ↓ | ACC ↑ | NLL ↓ | ECE ↓ |
| **Variational inference** | | | | | | |
| MC-Dropout | $79.50 \pm 0.37$ | $0.9162 \pm 0.0103$ | $0.0993 \pm 0.0033$ | $82.30 \pm 0.19$ | $0.6500 \pm 0.0049$ | $0.0574 \pm 0.0028$ |
| F-MC-Dropout | $\mathbf{81.06 \pm 0.44}$ | $\mathbf{0.7027 \pm 0.0049}$ | $\mathbf{0.0514 \pm 0.0047}$ | $\mathbf{83.24 \pm 0.11}$ | $\mathbf{0.6144 \pm 0.0068}$ | $\mathbf{0.0250 \pm 0.0027}$ |
| Deep-ens | $82.08 \pm 0.42$ | $0.7189 \pm 0.0108$ | $0.0334 \pm 0.0064$ | $83.04 \pm 0.15$ | $0.6958 \pm 0.0335$ | $0.0483 \pm 0.0017$ |
| F-Deep-ens | $\mathbf{82.54 \pm 0.10}$ | $\mathbf{0.6286 \pm 0.0022}$ | $\mathbf{0.0143 \pm 0.0041}$ | $\mathbf{84.52 \pm 0.03}$ | $\mathbf{0.5644 \pm 0.0106}$ | $\mathbf{0.0191 \pm 0.0039}$ |
| **Markov chain Monte Carlo** | | | | | | |
| SGLD | $80.13 \pm 0.01$ | $0.7604 \pm 0.0010$ | $0.1161 \pm 0.0031$ | $81.38 \pm 0.10$ | $0.7123 \pm 0.0204$ | $0.0958 \pm 0.0004$ |
| F-SGLD | $\mathbf{80.82 \pm 0.02}$ | $\mathbf{0.7276 \pm 0.0012}$ | $\mathbf{0.1085 \pm 0.0008}$ | $\mathbf{82.12 \pm 0.16}$ | $\mathbf{0.6722 \pm 0.0112}$ | $\mathbf{0.0820 \pm 0.0021}$ |
| **Sample** | | | | | | |
| SWAG-Diag | $80.18 \pm 0.50$ | $0.6837 \pm 0.0186$ | $\mathbf{0.0239 \pm 0.0047}$ | $82.40 \pm 0.09$ | $0.6150 \pm 0.0029$ | $0.0322 \pm 0.0018$ |
| F-SWAG-Diag | $\mathbf{81.01 \pm 0.29}$ | $\mathbf{0.6645 \pm 0.0050}$ | $0.0242 \pm 0.0039$ | $\mathbf{83.50 \pm 0.29}$ | $\mathbf{0.5763 \pm 0.0120}$ | $\mathbf{0.0151 \pm 0.0020}$ |
| SWAG | $79.90 \pm 0.50$ | $\mathbf{0.6595 \pm 0.0019}$ | $0.0587 \pm 0.0048$ | $82.23 \pm 0.19$ | $0.6078 \pm 0.0006$ | $\mathbf{0.0113 \pm 0.0020}$ |
| F-SWAG | $\mathbf{80.93 \pm 0.27}$ | $0.6704 \pm 0.0049$ | $\mathbf{0.0350 \pm 0.0025}$ | $\mathbf{83.57 \pm 0.26}$ | $\mathbf{0.5757 \pm 0.0136}$ | $0.0196 \pm 0.0015$ |

Table 2: Classification score on CIFAR-10 dataset. Each experiment is repeated three times with different random seeds and reports the mean and standard deviation.

| | PreResNet-164 | | | WideResNet28x10 | | |
|---|---|---|---|---|---|---|
| Method | ACC ↑ | NLL ↓ | ECE ↓ | ACC ↑ | NLL ↓ | ECE ↓ |
| **Variational inference** | | | | | | |
| MC-Dropout | $96.18 \pm 0.02$ | $0.1270 \pm 0.0030$ | $0.0162 \pm 0.0007$ | $96.39 \pm 0.09$ | $0.1094 \pm 0.0021$ | $\mathbf{0.0094 \pm 0.0014}$ |
| F-MC-Dropout | $\mathbf{96.39 \pm 0.18}$ | $\mathbf{0.1137 \pm 0.0024}$ | $\mathbf{0.0118 \pm 0.0006}$ | $\mathbf{97.10 \pm 0.12}$ | $\mathbf{0.0966 \pm 0.0047}$ | $0.0095 \pm 0.0008$ |
| Deep-ens | $96.39 \pm 0.09$ | $0.1277 \pm 0.0030$ | $0.0108 \pm 0.0015$ | $96.96 \pm 0.10$ | $0.1031 \pm 0.0076$ | $0.0087 \pm 0.0018$ |
| F-Deep-ens | $\mathbf{96.70 \pm 0.04}$ | $\mathbf{0.1031 \pm 0.0016}$ | $\mathbf{0.0057 \pm 0.0031}$ | $\mathbf{97.11 \pm 0.10}$ | $\mathbf{0.0851 \pm 0.0011}$ | $\mathbf{0.0059 \pm 0.0012}$ |
| **Markov chain Monte Carlo** | | | | | | |
| SGLD | $94.79 \pm 0.10$ | $0.2089 \pm 0.0021$ | $0.0711 \pm 0.0061$ | $95.87 \pm 0.08$ | $0.1573 \pm 0.0190$ | $0.0463 \pm 0.0050$ |
| F-SGLD | $\mathbf{95.04 \pm 0.06}$ | $\mathbf{0.1912 \pm 0.0080}$ | $\mathbf{0.0601 \pm 0.0002}$ | $\mathbf{96.43 \pm 0.05}$ | $\mathbf{0.1336 \pm 0.004}$ | $\mathbf{0.0385 \pm 0.0003}$ |
| **Sample** | | | | | | |
| SWAG-Diag | $96.03 \pm 0.10$ | $0.1251 \pm 0.0029$ | $0.0082 \pm 0.0008$ | $96.41 \pm 0.05$ | $0.1077 \pm 0.0009$ | $0.0047 \pm 0.0013$ |
| F-SWAG-Diag | $\mathbf{96.23 \pm 0.01}$ | $\mathbf{0.1108 \pm 0.0013}$ | $\mathbf{0.0043 \pm 0.0005}$ | $\mathbf{97.05 \pm 0.08}$ | $\mathbf{0.0888 \pm 0.0052}$ | $\mathbf{0.0043 \pm 0.0004}$ |
| SWAG | $96.03 \pm 0.02$ | $0.1232 \pm 0.0022$ | $\mathbf{0.0053 \pm 0.0004}$ | $96.32 \pm 0.08$ | $0.1122 \pm 0.0009$ | $0.0088 \pm 0.0006$ |
| F-SWAG | $\mathbf{96.25 \pm 0.03}$ | $\mathbf{0.11062 \pm 0.0014}$ | $0.0056 \pm 0.0002$ | $\mathbf{97.09 \pm 0.14}$ | $\mathbf{0.0883 \pm 0.0004}$ | $\mathbf{0.0036 \pm 0.0008}$ |

Finally, the objective function in (6) indicates that we aim to find an approximate posterior distribution that ensures any model sampled from it is aware of the sharpness, while also preferring simpler approximate posterior distributions. This preference can be estimated based on how we equip these distributions. With the Bayesian setting and variational inference formulation, our proposed sharpness-aware posterior can be integrated into MCMC-based and variational inference-based Bayesian Neural Networks. The supplementary material contains the details on how to derive variational approaches and incorporate the sharpness-awareness into the BNNs used in our experiments including BNNs with an approximate Gaussian distribution [33], BNNs with stochastic gradient Langevin dynamics (SGLD) [58], MC-Dropout [18], Bayesian deep ensemble [35], and SWAG [39].

## 4 Experiments

In this section, we conduct various experiments to demonstrate the effectiveness of the sharpness-aware approach on Bayesian Neural networks, including BNNs with an approximate Gaussian distribution [33] (i.e., SGVB for model's reparameterization trick and SGVB-LRT for representation's reparameterization trick), BNNs with stochastic gradient Langevin dynamics (SGLD) [58], MC-Dropout [18], Bayesian deep ensemble [35], and SWAG [39]. The experiments are conducted on three benchmark datasets: CIFAR-10, CIFAR-100, and ImageNet ILSVRC-2012, and report accuracy, negative log-likelihood (NLL), and Expected Calibration Error (ECE) to estimate the calibration capability and uncertainty of our method against baselines. The details of the dataset and implementation are described in the supplementary material[2].

---

[2]The implementation is provided in `https://github.com/anh-ntv/flat_bnn.git`

Table 3: Classification scores of approximate the Gaussian posterior on the CIFAR datasets. Each experiment is repeated three times with different random seeds and reports the mean and standard deviation.

| | **Resnet10** | | | **Resnet18** | | |
|---|---|---|---|---|---|---|
| Method | ACC ↑ | NLL ↓ | ECE ↓ | ACC ↑ | NLL ↓ | ECE ↓ |
| **Experiments on Cifar-100 dataset** | | | | | | |
| SGVB-LRT | $61.75 \pm 0.75$ | $1.534 \pm 0.03$ | $0.0676 \pm 0.01$ | $68.95 \pm 1.20$ | $1.140 \pm 0.21$ | $0.063 \pm 0.04$ |
| F-SGVB-LRT | $62.25 \pm 0.57$ | $1.4001 \pm 0.04$ | $0.0642 \pm 0.01$ | $70.00 \pm 1.42$ | $1.127 \pm 0.25$ | $\mathbf{0.022 \pm 0.05}$ |
| + Geometry | $\mathbf{62.54 \pm 0.67}$ | $\mathbf{1.3704 \pm 0.01}$ | $\mathbf{0.0301 \pm 0.03}$ | $\mathbf{70.12 \pm 1.02}$ | $\mathbf{1.121 \pm 0.23}$ | $0.036 \pm 0.06$ |
| SGVB | $54.40 \pm 0.98$ | $1.968 \pm 0.05$ | $0.214 \pm 0.00$ | $60.91 \pm 2.31$ | $1.746 \pm 0.15$ | $0.246 \pm 0.03$ |
| F-SGVB | $54.53 \pm 0.33$ | $1.967 \pm 0.00$ | $0.212 \pm 0.00$ | $61.54 \pm 2.23$ | $1.695 \pm 0.15$ | $0.242 \pm 0.03$ |
| + Geometry | $\mathbf{55.53 \pm 0.65}$ | $\mathbf{1.906 \pm 0.02}$ | $\mathbf{0.207 \pm 0.00}$ | $\mathbf{62.58 \pm 0.53}$ | $\mathbf{1.612 \pm 0.03}$ | $\mathbf{0.224 \pm 0.00}$ |
| **Experiments on Cifar-10 dataset** | | | | | | |
| SGVB-LRT | $84.98 \pm 1.87$ | $0.422 \pm 0.10$ | $0.043 \pm 0.04$ | $89.10 \pm 1.32$ | $0.344 \pm 0.02$ | $0.033 \pm 0.02$ |
| F-SGVB-LRT | $86.32 \pm 1.34$ | $0.409 \pm 0.03$ | $\mathbf{0.017 \pm 0.06}$ | $90.00 \pm 1.10$ | $0.291 \pm 0.02$ | $0.019 \pm 0.01$ |
| + Geometry | $\mathbf{86.44 \pm 1.12}$ | $\mathbf{0.403 \pm 0.06}$ | $0.025 \pm 0.03$ | $\mathbf{90.31 \pm 1.11}$ | $\mathbf{0.262 \pm 0.01}$ | $\mathbf{0.014 \pm 0.02}$ |
| SGVB | $80.52 \pm 2.10$ | $0.781 \pm 0.23$ | $0.237 \pm 0.06$ | $86.74 \pm 1.25$ | $0.541 \pm 0.01$ | $0.181 \pm 0.02$ |
| F-SGVB | $80.60 \pm 1.88$ | $0.776 \pm 0.13$ | $0.223 \pm 0.05$ | $\mathbf{87.01 \pm 0.91}$ | $0.534 \pm 0.01$ | $0.183 \pm 0.01$ |
| + Geometry | $\mathbf{82.05 \pm 0.47}$ | $\mathbf{0.704 \pm 0.01}$ | $\mathbf{0.206 \pm 0.00}$ | $86.80 \pm 1.30$ | $\mathbf{0.531 \pm 0.01}$ | $\mathbf{0.175 \pm 0.01}$ |

Table 4: Classification score on ImageNet dataset

| | **Densenet-161** | | | **ResNet-152** | | |
|---|---|---|---|---|---|---|
| Model | ACC ↑ | NLL ↓ | ECE ↓ | ACC ↑ | NLL ↓ | ECE ↓ |
| SWAG-Diag | 78.59 | 0.8559 | 0.0459 | 78.96 | 0.8584 | 0.0566 |
| F-SWAG-Diag | **78.71** | **0.8267** | **0.0194** | **79.20** | **0.8065** | **0.0199** |
| SWAG | 78.59 | 0.8303 | 0.0204 | 79.08 | 0.8205 | 0.0279 |
| F-SWAG | **78.70** | **0.8262** | **0.0185** | **79.17** | **0.8078** | **0.0208** |
| SGLD | 78.50 | 0.8317 | **0.0157** | 79.00 | 0.8165 | 0.0220 |
| F-SGLD | **78.64** | **0.8236** | 0.0166 | **79.16** | **0.8050** | **0.0167** |

## 4.1 Experimental results

### 4.1.1 Predictive performance

Our experimental results, presented in Tables 1, 2, 3 for CIFAR-100 and CIFAR-10 dataset, and Table 4 for the ImageNet dataset, indicate a notable improvement across all experiments. It is worth noting that there is a trade-off between accuracy, negative log-likelihood, and expected calibration error. Nonetheless, our approach obtains a fine balance between these factors compared to the overall improvement.

## 4.2 Effectiveness of sharpness-aware posterior

**Calibration of uncertainty estimates:** We evaluate the ECE of each setting and compare it to baselines in Tables 1, 2, and 4. This score measures the maximum discrepancy between the accuracy

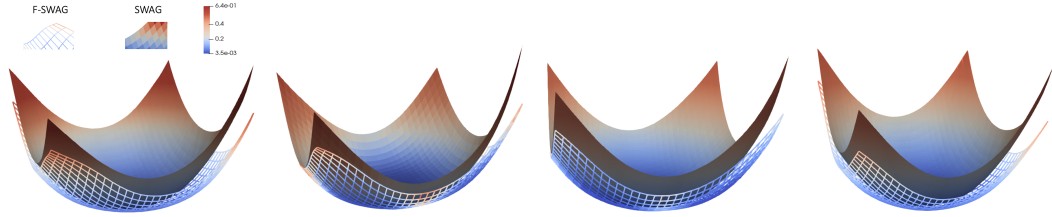

Figure 1: Comparing loss landscape of PreResNet-164 on CIFAR-100 dataset training with SWAG and F-SWAG method. For visualization purposes, we sample two models for each SWAG and F-SWAG and then plot the loss landscapes. It can be observed that the loss landscapes of our F-SWAG are flatter, supporting our argument for the flatter sampled models.

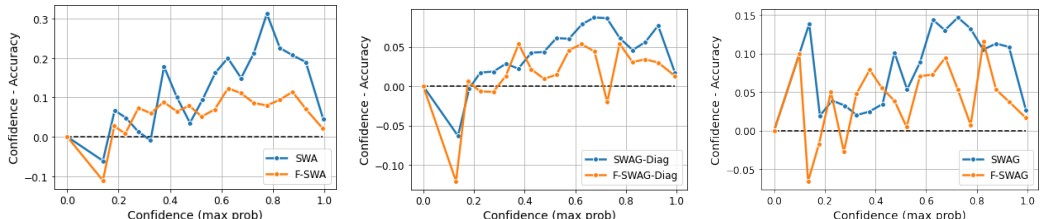

Figure 2: Reliability diagrams for PreResNet164 on CIFAR-100. The confidence is split into 20 bins and plots the gap between confidence and accuracy in each bin. The best case is the black dashed line when this gap is zeros. The plots of F-SWAG get closer to the zero lines, implying our F-SWAG can calibrate the uncertainty better.

Table 5: Classification score on CIFAR-10-C on PreResNet-164 model when training with CIFAR-10. The full result on each type of corruption is displayed in the supplemetary material.

| | ECE ↓ | | | | Accuracy ↑ | | | |
|---|---|---|---|---|---|---|---|---|
| Corruption | SWAG-D | F-SWAG-D | SWAG | F-SWAG | SWAG-D | F-SWAG-D | SWAG | F-SWAG |
| Noise | 0.0729 | 0.0701 | 0.0958 | 0.0078 | 74.26 | 75.59 | 74.02 | 75.08 |
| Blur | 0.0121 | 0.0090 | 0.0202 | 0.0273 | 91.13 | 90.55 | 91.03 | 90.93 |
| Weather | 0.018 | 0.0142 | 0.0272 | 0.0240 | 89.47 | 89.18 | 89.42 | 89.11 |
| Digital and others | 0.0277 | 0.0229 | 0.0384 | 0.0209 | 87.03 | 86.94 | 86.93 | 87.19 |
| Average | 0.0328 | **0.0290** | 0.0454 | **0.0200** | 85.47 | **85.56** | 85.35 | **85.58** |

and confidence of the model. To further clarify it, we display the Reliability Diagrams of PreResNet-164 on CIFAR-100 to understand how well the model predicts according to the confidence threshold in Figure 2. The experiments is detailed in the supplementary material.

**Out-of-distribution prediction:** The effectiveness of the sharpness-aware Bayesian neural network (BNN) is demonstrated in the above experiments, particularly in comparison to non-flat methods. In this section, we extend the evaluation to an out-of-distribution setting. Specifically, we utilize the BNN models trained on the CIFAR-10 dataset to assess their performance on the CIFAR-10-C dataset. This is an extension of the CIFAR-10 designed to evaluate the robustness of machine learning models against common corruptions and perturbations in the input data. The corruptions include various forms of noise, blur, weather conditions, and digital distortions. We conduct an ensemble of 30 models sampled from the flat-posterior distribution and compared them with non-flat ones. We present the average result of each corruption group and the average result on the whole dataset in Table 5, the detailed result of each corruption form is displayed in the supplementary material. Remarkably, the flat BNN models consistently surpass their non-flat counterparts with respect to average ECE and accuracy metrics. This finding is additional evidence of the generalization ability of the sharpness-aware posterior.

### 4.3 Ablation studies

In Figure 1, we plot the loss-landscape of the models sampled from our proposal of sharpness-aware posterior against the non-sharpness-aware one. Particularly, we compare two methods F-SWAG and SWAG by selecting four random models sampled from the posterior distribution of each method under the same hyper-parameter settings. As observed, our method not only improves the generalization of ensemble inference, demonstrated by classification results in Section 4.1 and sharpness in Section 4.2, but also the individual sampled model is flatter itself.

We measure and visualize the sharpness of the models. To this end, we sample five models from the approximate posteriors and then take the average of the sharpness of these models. For a model $\theta$, the sharpness is evaluated as $\max\limits_{||\epsilon||_2 \leq \rho} \mathcal{L}_\mathcal{S}(\theta + \epsilon) - \mathcal{L}_\mathcal{S}(\theta)$ to measure the change of loss value around $\theta$. We calculate the sharpness score of PreResNet-164 network for SWAG, and F-SWAG training on CIFAR-100 dataset and visualize them in the supplementary material. As shown there, the sharpness-aware versions produce smaller *sharpness* scores compared to the corresponding baselines, indicating that our models get into flatter regions.

# 5 Conclusion

In this paper, we introduce theories in the Bayesian setting and discuss variational inference for the sharpness-aware posterior in the context of Bayesian Neural Networks (BNNs). The sharpness-aware posterior results in models that are less sensitive to noise and have a better generalization ability, as it enables the models sampled from it and the optimal approximate posterior estimates to have a higher flatness. We conducted extensive experiments that leveraged the sharpness-aware posterior with state-of-the-art Bayesian Neural Networks. Our main results show that the models sampled from the proposed posterior outperform their baselines in terms of ensemble accuracy, expected calibration error (ECE), and negative log-likelihood (NLL). This indicates that the flat-seeking counterparts are better at capturing the true distribution of weights in neural networks and providing accurate probabilistic predictions. Furthermore, we performed ablation studies to showcase the effectiveness of the flat posterior distribution on various factors such as uncertainty estimation, loss landscape, and out-of-distribution prediction. Overall, the sharpness-aware posterior presents a promising approach for improving the generalization performance of Bayesian neural networks.

**Acknowledgements.** This work was partly supported by ARC DP23 grant DP230101176 and by the Air Force Office of Scientific Research under award number FA2386-23-1-4044.

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
