# Supplementary Material for Flat Seeking Bayesian Neural Networks

**Van-Anh Nguyen**[1]     **Tung-Long Vuong**[1]     **Hoang Phan**[2,3]     **Thanh-Toan Do**[1]

**Dinh Phung** [1,2]     **Trung Le** [1]

[1]Department of Data Science and AI, Monash University, Australia

[2]VinAI, Vietnam

[3]New York University, United States

{van-anh.nguyen, tung-long.vuong, toan.do, dinh.phung, trunglm}@monash.edu

hvp2011@nyu.edu

## 1  Theoretical Development

### 1.1  All Proofs

**Theorem 1.1.** *(Theorem 3.1 in the main paper) Consider the following optimization problem*

$$\min_{\mathbb{Q}<<\mathbb{P}} \left\{ \lambda \mathcal{L}_S (\mathbb{Q}) + KL (\mathbb{Q}, \mathbb{P}) \right\}, \tag{1}$$

*where we search over $\mathbb{Q}$ absolutely continuous w.r.t. $\mathbb{P}$ and $KL (\cdot, \cdot)$ is the Kullback-Leibler divergence. This optimization has a closed-form optimal solution $\mathbb{Q}^*$ with the density*

$$q^* (\theta) \propto \exp \left\{ -\lambda \mathcal{L}_S (\theta) \right\} p(\theta),$$

*which is exactly the standard posterior $\mathbb{Q}_S$ with the density $q(\theta \mid \mathcal{S})$.*

*Proof.* We have

$$\lambda \mathcal{L}_S (\mathbb{Q}) + KL (\mathbb{Q}, \mathbb{P}) = \lambda \int \mathcal{L}_S (\theta) q (\theta) \, d\theta + \int q(\theta) \log \frac{q (\theta)}{p (\theta)} d\theta.$$

The Lagrange function is as follows

$$L (q, \alpha) = \lambda \int \mathcal{L}_S (\theta) q (\theta) \, d\theta + \int q(\theta) \log \frac{q (\theta)}{p (\theta)} d\theta + \alpha \left( \int q(\theta) d\theta - 1 \right).$$

Take derivative w.r.t. $q (\theta)$ and set it to 0, we obtain

$$\lambda \mathcal{L}_S (\theta) + \log q (\theta) + 1 - \log p (\theta) + \alpha = 0.$$

$$q (\theta) = \exp \left\{ -\lambda \mathcal{L}_S (\theta) \right\} p (\theta) \exp \left\{ -\alpha - 1 \right\}.$$

$$q (\theta) \propto \exp \left\{ -\lambda \mathcal{L}_S (\theta) \right\} p (\theta).$$

$\square$

**Lemma 1.2.** *Assume that the data space $\mathcal{X}$, the label space $\mathcal{Y}$, and the model space $\Theta$ are compact sets. There exist the modulus of continuity $\omega : \mathbb{R}^+ \to \mathbb{R}^+$ with $\lim_{t \to 0^+} \omega (t) = 0$ such that $|\ell (f_\theta (x), y) - \ell (f_{\theta'} (x), y)| \leq \omega (\|\theta - \theta'\|), \forall x \in \mathcal{X}, y \in \mathcal{Y}.$*

37th Conference on Neural Information Processing Systems (NeurIPS 2023).

*Proof.* The loss function $\ell(f_\theta(x), y)$ is continuous on the compact set $\mathcal{X} \times \mathcal{Y} \times \Theta$, hence it is equip-continuous on this set. For every $\epsilon > 0$, there exists $\delta_x, \delta_y, \delta_\theta > 0$ such that

$$\forall \|x' - x\| \leq \delta_x, \|y' - y\| \leq \delta_y, \|\theta' - \theta\| \leq \delta_\theta,$$

we have $|\ell(f_{\theta'}(x'), y') - \ell(f_{\theta'}(x), y)| \leq \epsilon$.

Therefore, for all $\|\theta' - \theta\| \leq \delta_\theta$, we have

$$|\ell(f_\theta(x), y) - \ell(f_{\theta'}(x), y)| \leq \epsilon, \forall x, y.$$

This means that the family $\{\ell(f_\theta(x), y) : x \in \mathcal{X}, y \in \mathcal{Y}\}$ is equi-continuous w.r.t. $\theta \in \Theta$. This means the existence of the common modulus of continuity $\omega : \mathbb{R}^+ \to \mathbb{R}^+$ with $\lim_{t \to 0^+} \omega(t) = 0$. $\square$

**Definition 1.3.** Given $\epsilon > 0$, we say that $\Theta$ is $\epsilon$-covered by a set $\Theta'$ if for all $\theta \in \Theta$, there exists $\theta' \in \Theta'$ such that $\|\theta' - \theta\| \leq \epsilon$. We define $\mathcal{N}(\Theta, \epsilon)$ as the cardinality set of the smallest set $\Theta'$ that covers $\Theta$.

**Lemma 1.4.** *Let $R = \max_{\theta \in \Theta} \|\theta\|^2 < \infty$ and $k$ is the dimension of $\Theta$. We can upper-bound the coverage number as*

$$\mathcal{N}(\Theta, \epsilon) \leq \left(\frac{2R\sqrt{k}}{\epsilon}\right)^k.$$

*Proof.* The proof can be found in Chapter 27 of [6]. $\square$

By choosing $\epsilon = \frac{1}{n^{\frac{1}{2k}}}$, we obtain

$$\mathcal{N}\left(\Theta, n^{-\frac{1}{2k}}\right) \leq \left(2R\sqrt{k}\right)^k \sqrt{n}.$$

However, solving the optimization problem (OP) for the general data-label distribution $\mathcal{D}$ is generally intractable. To make it tractable, we find its upper-bound which is relevant to the sharpness as shown in the following theorem.

**Theorem 1.5.** *(Theorem 3.2 in the main paper) Assume that $\Theta$ is a compact set. Given any $\delta \in [0; 1]$, with the probability at least $1 - \delta$ over the choice of $\mathcal{S} \sim \mathcal{D}^n$, for any distribution $\mathbb{Q}$, we have*

$$\mathcal{L}_\mathcal{D}(\mathbb{Q}) \leq \mathbb{E}_{\theta \sim \mathbb{Q}}\left[\max_{\theta' : \|\theta' - \theta\| \leq \rho} \mathcal{L}_\mathcal{S}(\theta')\right] + \mathcal{L}_\mathcal{S}(\mathbb{Q}) + \frac{1}{\sqrt{n}} + 2\omega\left(\frac{1}{n^{\frac{1}{2k}}}\right)$$

$$+ \sqrt{\frac{k\left(1 + \log\left(1 + \frac{2R^2}{\rho^2}\left(1 + 2\log\left(2R\sqrt{k}\right) + \frac{2}{k}\log n\right)\right)\right) + 2\log\frac{n}{\delta}}{4(n-1)}},$$

*where we assume that $\mathcal{L}_\mathcal{D}(\mathbb{Q}) = \mathbb{E}_{\theta \sim \mathbb{Q}}[\mathcal{L}_\mathcal{D}(\theta)] \leq \mathbb{E}_{\theta \sim \mathbb{Q}}[\mathbb{E}_{\epsilon \sim \mathcal{N}(0, \sigma\mathbb{I})}[\mathcal{L}_\mathcal{D}(\theta + \epsilon)]]$ with $\sigma = \frac{\rho}{k^{1/2}\left(1 + \sqrt{\frac{\log(N^2 n)}{k}}\right)}$ and $N = \mathcal{N}\left(\Theta, n^{-\frac{1}{2k}}\right)$, $k$ is the number of parameters of the models, $n = |S|$, $R = \max_{\theta \in \Theta} \|\theta\|$, and $\omega : \mathbb{R}^+ \to \mathbb{R}^+$ is a function such that $\lim_{t \to 0^+} \omega(t) = 0$.*

*Proof.* Given $\epsilon = \frac{1}{n^{\frac{1}{2k}}}$, we denote $\Theta' = \left\{\theta'_1, \ldots, \theta'_N\right\}$ where $N = \mathcal{N}\left(\Theta, n^{-\frac{1}{2k}}\right) \leq \left(2R\sqrt{k}\right)^k \sqrt{n}$ as the $\epsilon$-covered set of $\Theta$. We first examine a discrete distribution

$$\mathbb{Q} = \sum_{i=1}^m \pi_i \delta_{\theta_i}.$$

Without lossing the generalization, we can assume that $\left\|\theta'_i - \theta_i\right\| \leq \epsilon, \forall i = 1, \ldots, m$. We note that $\theta'_1, \ldots \theta'_m$ can be repeated if $m > N$. Using Lemma 1.2, let $\omega(\cdot)$ be the modulus of continuity

of $\ell(f_\theta(x), y)$ such that $|\ell(f_\theta(x), y) - \ell(f_\theta(x), y)| \leq \omega(\|\theta - \theta'\|), \forall x, y$ and $\lim_{t \to 0} \omega(t) = 0$. This implies that

$$\left| \ell(f_{\theta_i}(x), y) - \ell\left(f_{\theta'_i}(x), y\right) \right| \leq \omega(\epsilon) = \omega\left(\frac{1}{n^{\frac{1}{2k}}}\right), \forall x, y, i = 1, \dots, m.$$

We consider the distribution $\bar{\mathbb{Q}} = \sum_{i=1}^m \pi_i \mathcal{N}\left(\theta'_i, \sigma\mathbb{I}\right)$. According to the McAllester PAC-Bayes bound, with the probability $1 - \delta$ over the choices of $\mathcal{S} \sim \mathcal{D}^n$, for any distribution $\bar{\mathbb{P}}$, we have

$$\mathcal{L}_\mathcal{D}\left(\bar{\mathbb{Q}}\right) \leq \mathcal{L}_\mathcal{S}\left(\bar{\mathbb{Q}}\right) + \sqrt{\frac{KL\left(\bar{\mathbb{Q}}, \bar{\mathbb{P}}\right) + \log\frac{n}{\delta}}{2(n-1)}}.$$

Let $\theta^* = \underset{1 \leq i \leq m}{\text{argmax}} \left\|\theta'_i\right\|$. We consider the distribution $\bar{\mathbb{P}} = \mathcal{N}(0, \sigma_\mathbb{P})$ where $\sigma_\mathbb{P}^2 = c\exp\left\{\frac{1-j}{k}\right\}$ with $c = \sigma^2\left(1 + \exp\left\{\frac{4n}{k}\right\}\right)$ and $j = \left\lfloor 1 + k\log\frac{c}{\sigma^2 + \frac{\|\theta^*\|^2}{k}} \right\rfloor = \left\lfloor 1 + k\log\frac{\sigma^2\left(1 + \exp\left\{\frac{4n}{k}\right\}\right)}{\sigma^2 + \frac{\|\theta^*\|^2}{k}} \right\rfloor$. It follows that

$$\sigma^2 + \frac{\|\theta^*\|^2}{k} \leq \sigma_\mathbb{P} \leq \exp\left\{\frac{1}{k}\right\}\left(\sigma^2 + \frac{\|\theta^*\|^2}{k}\right).$$

We have

$$KL\left(\mathcal{N}\left(\theta'_i, \sigma\mathbb{I}\right), \bar{\mathbb{P}}\right) = \frac{1}{2}\left[\frac{k\sigma^2 + \left\|\theta'_i\right\|^2}{\sigma_\mathbb{P}^2} - k + k\log\left(\frac{\sigma_\mathbb{P}^2}{\sigma^2}\right)\right].$$

$$KL\left(\mathcal{N}\left(\theta^*, \sigma\mathbb{I}\right), \bar{\mathbb{P}}\right) = \max_i KL\left(\mathcal{N}\left(\theta'_i, \sigma\mathbb{I}\right), \bar{\mathbb{P}}\right).$$

$$KL\left(\bar{\mathbb{Q}}, \bar{\mathbb{P}}\right) \leq \sum_{i=1}^m \pi_i KL\left(\mathcal{N}\left(\theta'_i, \sigma\mathbb{I}\right), \bar{\mathbb{P}}\right) \leq KL\left(\mathcal{N}\left(\theta^*, \sigma\mathbb{I}\right), \bar{\mathbb{P}}\right).$$

We now bound $KL\left(\mathcal{N}\left(\theta^*, \sigma\mathbb{I}\right), \bar{\mathbb{P}}\right)$

$$KL\left(\mathcal{N}\left(\theta^*, \sigma\mathbb{I}\right), \bar{\mathbb{P}}\right) = \frac{1}{2}\left[\frac{k\sigma^2 + \|\theta^*\|^2}{\sigma_\mathbb{P}^2} - k + k\log\left(\frac{\sigma_\mathbb{P}^2}{\sigma^2}\right)\right]$$

$$\leq \frac{1}{2}\left[\frac{k\sigma^2 + \|\theta^*\|^2}{\sigma^2 + \frac{\|\theta^*\|^2}{k}} - k + k\log\left(\frac{\exp\left\{\frac{1}{k}\right\}\left(\sigma^2 + \frac{\|\theta^*\|^2}{k}\right)}{\sigma^2}\right)\right]$$

$$\leq \frac{k}{2}\left(1 + \log\left(1 + \frac{\|\theta^*\|^2}{k\sigma^2}\right)\right).$$

Therefore, with the probability $1 - \delta$, we reach

$$\mathcal{L}_\mathcal{D}\left(\bar{\mathbb{Q}}\right) \leq \mathcal{L}_\mathcal{S}\left(\bar{\mathbb{Q}}\right) + \sqrt{\frac{k\left(1 + \log\left(1 + \frac{\|\theta^*\|^2}{k\sigma^2}\right)\right) + 2\log\frac{n}{\delta}}{4(n-1)}}.$$

$$\mathbb{E}_{\theta \sim \sum_{i=1}^m \pi_i \mathcal{N}(\theta'_i, \sigma\mathbb{I})}\left[\mathcal{L}_\mathcal{D}(\theta)\right] \leq \mathbb{E}_{\theta \sim \sum_{i=1}^m \pi_i \mathcal{N}(\theta'_i, \sigma\mathbb{I})}\left[\mathcal{L}_\mathcal{S}(\theta)\right] + \sqrt{\frac{k\left(1 + \log\left(1 + \frac{\|\theta^*\|^2}{k\sigma^2}\right)\right) + 2\log\frac{n}{\delta}}{4(n-1)}}.$$

$$\leq \sum_{i=1}^m \pi_i \mathbb{E}_{\epsilon_i \sim \mathcal{N}(0, \sigma\mathbb{I})}\left[\mathcal{L}_\mathcal{S}\left(\theta'_i + \epsilon_i\right)\right] + \sqrt{\frac{k\left(1 + \log\left(1 + \frac{\|\theta^*\|^2}{k\sigma^2}\right)\right) + 2\log\frac{n}{\delta}}{4(n-1)}}.$$

Note that

$$\mathbb{E}_{\theta \sim \mathcal{N}(\theta_i, \sigma\mathbb{I})}\left[\mathcal{L}_{\mathcal{D}}(\theta)\right] - \mathbb{E}_{\theta \sim \mathcal{N}(\theta'_i, \sigma\mathbb{I})}\left[\mathcal{L}_{\mathcal{D}}(\theta)\right] = \int \left[\mathcal{L}_{\mathcal{D}}(\theta_i + \epsilon_i) - \mathcal{L}_{\mathcal{D}}\left(\theta'_i + \epsilon_i\right)\right]\mathcal{N}(\epsilon_i \mid 0, \sigma I)\, d\epsilon_i$$

$$\leq \int \omega\left(\frac{1}{n^{\frac{1}{2k}}}\right)\mathcal{N}(\epsilon_i \mid 0, \sigma I)\, d\epsilon_i = \omega\left(\frac{1}{n^{\frac{1}{2k}}}\right).$$

$$\mathbb{E}_{\theta \sim \mathcal{N}(\theta_i, \sigma\mathbb{I})}\left[\mathcal{L}_{\mathcal{D}}(\theta)\right] \leq \mathbb{E}_{\theta \sim \mathcal{N}(\theta'_i, \sigma\mathbb{I})}\left[\mathcal{L}_{\mathcal{D}}(\theta)\right] + \omega\left(\frac{1}{n^{\frac{1}{2k}}}\right).$$

$$\sum_{i=1}^{m} \pi_i \mathbb{E}_{\theta \sim \mathcal{N}(\theta_i, \sigma\mathbb{I})}\left[\mathcal{L}_{\mathcal{D}}(\theta)\right] \leq \sum_{i=1}^{m} \pi_i \mathbb{E}_{\theta \sim \mathcal{N}(\theta'_i, \sigma\mathbb{I})}\left[\mathcal{L}_{\mathcal{D}}(\theta)\right] + \omega\left(\frac{1}{n^{\frac{1}{2k}}}\right),$$

therefore we have

$$\mathbb{E}_{\theta \sim \sum_{i=1}^{m} \pi_i \mathcal{N}(\theta_i, \sigma\mathbb{I})}\left[\mathcal{L}_{\mathcal{D}}(\theta)\right] \leq \sum_{i=1}^{m} \pi_i \mathbb{E}_{\epsilon_i \sim \mathcal{N}(0, \sigma\mathbb{I})}\left[\mathcal{L}_{\mathcal{S}}\left(\theta'_i + \epsilon_i\right)\right]$$

$$+ \sqrt{\frac{k\left(1 + \log\left(1 + \frac{\|\theta^*\|^2}{k\sigma^2}\right)\right) + 2\log\frac{n}{\delta}}{4(n-1)}} + \omega\left(\frac{1}{n^{\frac{1}{2k}}}\right).$$

Using the assumption

$$\mathcal{L}_{\mathcal{D}}(\mathbb{Q}) = \mathbb{E}_{\theta \sim \mathbb{Q}}\left[\mathcal{L}_{\mathcal{D}}(\theta)\right] \leq \mathbb{E}_{\theta \sim \mathbb{Q}}\left[\mathbb{E}_{\epsilon \sim \mathcal{N}(0, \sigma\mathbb{I})}\left[\mathcal{L}_{\mathcal{D}}(\theta + \epsilon)\right]\right] = \mathbb{E}_{\theta \sim \sum_{i=1}^{m} \pi_i \mathcal{N}(\theta_i, \sigma\mathbb{I})}\left[\mathcal{L}_{\mathcal{D}}(\theta)\right],$$

we obtain

$$\mathcal{L}_{\mathcal{D}}(\mathbb{Q}) \leq \sum_{i=1}^{m} \pi_i \mathbb{E}_{\epsilon_i \sim \mathcal{N}(0, \sigma\mathbb{I})}\left[\mathcal{L}_{\mathcal{S}}(\theta_i + \epsilon_i)\right]$$

$$+ \sqrt{\frac{k\left(1 + \log\left(1 + \frac{R^2}{k\sigma^2}\right)\right) + 2\log\frac{n}{\delta}}{4(n-1)}} + \omega\left(\frac{1}{n^{\frac{1}{2k}}}\right).$$

Because $\epsilon_i \sim \mathcal{N}(0, \sigma\mathbb{I})$, $\|\epsilon_i\|^2$ follows the Chi-squared distribution. Therefore, we have for any $i \in [m]$

$$\mathbb{P}\left(\|\epsilon_i\|^2 - k\sigma^2 \geq 2\sigma^2\sqrt{kt} + 2t\sigma^2\right) \leq \exp(-t), \forall t.$$

$$\mathbb{P}\left(\max_{i \in [m]}\|\epsilon_i\|^2 - k\sigma^2 \geq 2\sigma^2\sqrt{kt} + 2t\sigma^2\right) \leq N\exp(-t), \forall t.,$$

since the cardinality of $\left|\{\theta'_1, \ldots, \theta'_m\}\right|$ cannot exceed $N$.

$$\mathbb{P}\left(\max_{i \in [m]}\|\epsilon_i\|^2 - k\sigma^2 < 2\sigma^2\sqrt{kt} + 2t\sigma^2\right) > 1 - N\exp(-t), \forall t.$$

By choosing $t = \log\left(Nn^{1/2}\right)$, with the probability at least $1 - \frac{1}{\sqrt{n}}$, we have for all $i \in [m]$

$$\|\epsilon_i\|^2 < \sigma^2 k\left(1 + \frac{\log\left(N^2 n\right)}{k} + 2\sqrt{\frac{\log\left(Nn^{1/2}\right)}{k}}\right) \leq \sigma^2 k\left(1 + \sqrt{\frac{\log\left(N^2 n\right)}{k}}\right)^2.$$

By choosing $\sigma = \frac{\rho}{k^{1/2}\left(1 + \sqrt{\frac{\log\left(N^2 n\right)}{k}}\right)}$, with the probability at least $1 - \frac{1}{\sqrt{n}}$, we have for all $i \in [m]$

$$\|\epsilon_i\| < \rho.$$

We now derive

$$
\begin{aligned}
\mathcal{L}_{\mathcal{D}}\left(\mathbb{Q}\right) \leq & \sum_{i=1}^{m} \pi_i \left( \left( 1 - \frac{1}{\sqrt{n}} \right) \max_{\|\epsilon_i\|\leq\rho} \mathcal{L}_{\mathcal{S}} \left( \theta_i^{'} + \epsilon_i \right) \right. \\
& \left. + \frac{1}{\sqrt{n}} + \sqrt{ \frac{ k \left( 1 + \log \left( 1 + \frac{R^2}{k\sigma^2} \right) \right) + 2\log\frac{n}{\delta} }{4(n-1)} } + \omega \left( \frac{1}{n^{\frac{1}{2k}}} \right) \right) \\
\leq & \left( 1 - \frac{1}{\sqrt{n}} \right) \sum_{i=1}^{m} \pi_i \max_{\|\epsilon_i\|\leq\rho} \mathcal{L}_{\mathcal{S}} \left( \theta_i^{'} + \epsilon_i \right) + \frac{1}{\sqrt{n}} \\
& + \sqrt{ \frac{ k \left( 1 + \log \left( 1 + \frac{R^2}{\rho^2} \left( 1 + \sqrt{\frac{\log(N^2 n)}{k}} \right)^2 \right) \right) + 2\log\frac{n}{\delta} }{4(n-1)} } + \omega \left( \frac{1}{n^{\frac{1}{2k}}} \right) \\
\leq & \sum_{i=1}^{m} \pi_i \max_{\|\epsilon_i\|\leq\rho} \mathcal{L}_{\mathcal{S}} \left( \theta_i^{'} + \epsilon_i \right) + \frac{1}{\sqrt{n}} \\
& + \sqrt{ \frac{ k \left( 1 + \log \left( 1 + \frac{2R^2}{\rho^2} \left( 1 + \frac{\log(N^2 n)}{k} \right) \right) \right) + 2\log\frac{n}{\delta} }{4(n-1)} } + \omega \left( \frac{1}{n^{\frac{1}{2k}}} \right) \\
\leq & \sum_{i=1}^{m} \pi_i \max_{\|\epsilon_i\|\leq\rho} \mathcal{L}_{\mathcal{S}} \left( \theta_i^{'} + \epsilon_i \right) + \frac{1}{\sqrt{n}} \\
& + \sqrt{ \frac{ k \left( 1 + \log \left( 1 + \frac{2R^2}{\rho^2} \left( 1 + 2\log\left( 2R\sqrt{k} \right) + \frac{2}{k}\log n \right) \right) \right) + 2\log\frac{n}{\delta} }{4(n-1)} } + \omega \left( \frac{1}{n^{\frac{1}{2k}}} \right).
\end{aligned}
$$

Note that for all $i \in [m]$

$$
\max_{\|\epsilon_i\|\leq\rho} \mathcal{L}_{\mathcal{S}} \left( \theta_i^{'} + \epsilon_i \right) \leq \max_{\|\epsilon_i\|\leq\rho} \mathcal{L}_{\mathcal{S}} \left( \theta_i + \epsilon_i \right) + \omega \left( \frac{1}{n^{\frac{1}{2k}}} \right),
$$

therefore, we reach

$$
\begin{aligned}
\mathcal{L}_{\mathcal{D}}\left(\mathbb{Q}\right) \leq & \sum_{i=1}^{m} \pi_i \max_{\|\epsilon_i\|\leq\rho} \mathcal{L}_{\mathcal{S}} \left( \theta_i + \epsilon_i \right) + \frac{1}{\sqrt{n}} \\
& + \sqrt{ \frac{ k \left( 1 + \log \left( 1 + \frac{2R^2}{\rho^2} \left( 1 + 2\log\left( 2R\sqrt{k} \right) + \frac{2}{k}\log n \right) \right) \right) + 2\log\frac{n}{\delta} }{4(n-1)} } + 2\omega \left( \frac{1}{n^{\frac{1}{2k}}} \right) \\
\leq & \, \mathcal{L}_{\mathcal{S}}\left(\mathbb{Q}\right) + \frac{1}{\sqrt{n}} + 2\omega \left( \frac{1}{n^{\frac{1}{2k}}} \right) \\
& + \sqrt{ \frac{ k \left( 1 + \log \left( 1 + \frac{2R^2}{\rho^2} \left( 1 + 2\log\left( 2R\sqrt{k} \right) + \frac{2}{k}\log n \right) \right) \right) + 2\log\frac{n}{\delta} }{4(n-1)} }.
\end{aligned}
$$

For any distribution $\mathbb{Q}$, we approximate $\mathbb{Q}$ by its empirical distribution

$$
\mathbb{Q}_m = \frac{1}{m} \sum_{i=1}^{m} \delta_{\theta_i},
$$

which weakly converges to $\mathbb{Q}$ when $m \to \infty$. By using the achieved results for $\mathbb{Q}_m$ and taking limitation when $m \to \infty$, we reach the conclusion. $\qquad\square$

**Theorem 1.6.** *(Theorem 3.3 in the main paper) The optimal solution the OP in is the sharpness-aware posterior distribution* $\mathbb{Q}_S^{SA}$ *with the density function* $q^{SA}(\theta|\mathcal{S})$:

$$q^{SA}(\theta|\mathcal{S}) \propto \exp\left\{-\lambda \max_{\theta':\|\theta'-\theta\|\leq\rho} \mathcal{L}_{\mathcal{S}}(\theta')\right\} p(\theta)$$

$$= \exp\left\{-\lambda \mathcal{L}_{\mathcal{S}}(s(\theta))\right\} p(\theta),$$

*where we have defined* $s(\theta) = \underset{\theta':\|\theta'-\theta\|\leq\rho}{argmax} \mathcal{L}_{\mathcal{S}}(\theta')$.

*Proof.* We have

$$\lambda\mathbb{E}_{\theta\sim\mathbb{Q}}\left[\max_{\theta':\|\theta'-\theta\|\leq\rho}\mathcal{L}_{\mathcal{S}}(\theta')\right] + KL(\mathbb{Q},\mathbb{P}) = \lambda\int\mathcal{L}_S(s(\theta))q(\theta)d\theta + \int q(\theta)\log\frac{q(\theta)}{p(\theta)}d\theta.$$

The Lagrange function is as follows

$$L(q,\alpha) = \lambda\int\mathcal{L}_S(s(\theta))q(\theta)d\theta + \int q(\theta)\log\frac{q(\theta)}{p(\theta)}d\theta + \alpha\left(\int q(\theta)d\theta - 1\right).$$

Take derivative w.r.t. $q(\theta)$ and set it to 0, we obtain

$$\lambda\mathcal{L}_S(s(\theta)) + \log q(\theta) + 1 - \log p(\theta) + \alpha = 0.$$

$$q(\theta) = \exp\left\{-\lambda\mathcal{L}_S(s(\theta))\right\} p(\theta)\exp\left\{-\alpha - 1\right\}.$$

$$q(\theta) \propto \exp\left\{-\lambda\mathcal{L}_S(s(\theta))\right\} p(\theta).$$

$\square$

## 1.2 Technicalities of the baselines and the corresponding flat versions

In what follows, we present how the baselines used in the experiments can be viewed as variational and MCMC approaches and incorporate our sharpness-aware technique.

**Bayesian deep ensemble [3]:** We consider the approximate posterior $q_\phi = \frac{1}{K}\sum_{k=1}^{K}\delta_{\theta_k}$ where $\delta$ is the Dirac delta distribution as a uniform distribution over several base models $\theta_{1:K}$. Considering the prior distribution $p(\theta) = \mathcal{N}(0,\mathbb{I})$, we have the following OPs for the non-flat and flat versions.

**Non-flat version:**

$$\min_{\theta_{1:K}}\left\{\mathbb{E}_{\theta_k\sim q_\phi}\left[\lambda\mathcal{L}_S(\theta_k)\right] + KL\left(\frac{1}{K}\sum_{k=1}^{K}\delta_{\theta_k}, \mathcal{N}(0,\mathbb{I})\right)\right\},$$

where $KL\left(\frac{1}{K}\sum_{k=1}^{K}\delta_{\theta_k}, \mathcal{N}(0,\mathbb{I})\right) = -\frac{1}{K}\sum_{k=1}^{K}\log\mathcal{N}(\theta_k \mid 0,\mathbb{I}) + \text{const}$, leading to the L2 regularization terms.

**Flat version:**

$$\min_{\theta_{1:K}}\left\{\mathbb{E}_{\theta_k\sim q_\phi}\left[\lambda\max_{\theta':\|\theta'-\theta_k\|\leq\rho}\mathcal{L}_S(\theta')\right] + KL\left(\frac{1}{K}\sum_{k=1}^{K}\delta_{\theta_k}, \mathcal{N}(0,\mathbb{I})\right)\right\}.$$

**MC-Dropout [2]:** As shown in [2], the MC-dropout can be viewed as a BNN with the approximate posterior $q_\phi = \delta_\phi$ where $\phi$ is a fully-connected base model without any dropout and the prior distribution $p(\theta) = \mathcal{N}(0,\mathbb{I})$. The $KL(q_\phi, p(\theta))$ can be approximated which turns out to be a weighted L2 regularization where the weights are proportional to the keep-prob rates at the layers. The main term $\mathbb{E}_{\theta\sim q_\phi}\left[\lambda\mathcal{L}_S(\theta)\right]$ can be interpreted as applying the dropout before minimizing the loss. For our flat version, the main term is $\mathbb{E}_{\theta\sim q_\phi}\left[\lambda\max_{\theta':\|\theta'-\theta\|\leq\rho}\mathcal{L}_S(\theta')\right]$.

**BNNs with Stochastic Gradient Langevin Dynamics (SGLD) [7]:** For SGLD, we sample one or several particle models directly from the posterior distribution $q(\theta \mid S)$ for the non-flat version and from the SA-posterior distribution $q^{SA}(\theta \mid S)$ for the flat version. For the non-flat version, the update is similar to the mini-batch SGD except that we add small Gaussian noises to the particle models. For our flat version, we first compute the perturbed model $\theta^a$ for a given particle model $\theta$ and use the mini-batch SGD update with the gradient evaluated at $\theta^a$ together with small Gaussian noises.

**SWAG [4]:** We consider SWAG as an MCMC approach, where we keep a trajectory of particle models using SWA. Additionally, the covariance matrices are determined based on this trajectory to form an approximate Gaussian posterior. In the corresponding flat version to this approach, we employ SWA to sample from the SA (Sharpness-Aware) posterior. Specifically, we first calculate the perturbed model $\theta^a$ based on the current model $\theta$ and then employ mini-batch SGD updates with the gradient evaluated at model $\theta^a$. Finally, we update the final model using the SWA strategy.

## 2 Additional experiments

### 2.1 Comparison with bSAM method

We conduct experiments to compare our flat BNN with bSAM [5] on Resnet18, the results are shown in Table 1. The authors of bSAM explored the relationship between SAM and BNN and proposed a combination of SAM and Adam to optimize the mean of parameters in BNN networks while keeping the variance fixed. The results clearly indicate that our flat BNN outperforms bSAM in most metric scores. Here we note that we are unable to evaluate bSAM on the architectures used in Tables 1 and 2 in the main paper because the authors did not release the code. Instead, we run our methods with the setting mentioned in the bSAM paper.

Table 1: Classification score on Resnet18

| Method | CIFAR-10 | | | CIFAR-100 | | |
|---|---|---|---|---|---|---|
| | ACC ↑ | NLL ↓ | ECE ↓ | ACC ↑ | NLL ↓ | ECE ↓ |
| bSAM | 96.15 | 0.1200 | 0.0049 | 80.22 | **0.7000** | 0.0310 |
| F-SWAG-Diag | 96.56 | 0.1047 | **0.0037** | 80.70 | 0.7012 | **0.0227** |
| F-SWAG | **96.58** | **0.1045** | 0.0045 | **80.74** | 0.7024 | 0.0243 |

### 2.2 Full result of Out-of-distribution prediction

In Section 4.2 of the main paper, we provide a comprehensive analysis of the performance concerning various corruption groups, including noise, blur, weather conditions, and digital distortions. We present the detailed results for each corruption type in Table 2, providing a deeper understanding of the impact of these corruptions on the model's performance. On average, flat BNNs outperform their non-flat counterparts, especially on ECE with a notable margin. These findings further emphasize the effectiveness of flat BNNs in enhancing robustness and generalization against various corruptions.

### 2.3 Additional ablation studies

**Comparison of Hessian eigenvalue** We report the log scale of the largest eigenvalue of the Hessian matrix over several methods applying to WideResNet28x10 using CIFAR-100, and the ratio of the largest and fifth eigenvalue as shown in Table 3, which evidently indicates that our method updates models to minima having lower curvature.

**Computational cost** Our flat-seeking method requires the computation of gradients twice: initially to obtain the perturbed model $\theta'$ and subsequently to update the model. Consequently, the training time is nearly double in comparison to non-flat counterparts, as indicated in Table 4. Note that the Deep Ensemble settings utilize multiple models training individually for prediction and we report training time for one model in each setting.

**The effect of $KL$ term in Deep-ensemble settings** We present the results of training Deep-ensemble with SAM following the formula for the flat version in Section 1.2 but without KL (or L2 regularisation) in Table 5. Each experiment is performed three times and reports the mean and standard

Table 2: Classification score on CIFAR-10-C using PreResNet-164 model when training with CIFAR-10 dataset

| Method | ECE $\downarrow$ | | | | Accuracy $\uparrow$ | | | |
| --- | --- | --- | --- | --- | --- | --- | --- | --- |
| | SWAG-D | F-SWAG-D | SWAG | F-SWAG | SWAG-D | F-SWAG-D | SWAG | F-SWAG |
| Gaussian noise | 0.0765 | 0.0765 | 0.1032 | 0.0091 | 72.01 | 73.95 | 71.58 | 73.43 |
| Shot noise | 0.0661 | 0.0647 | 0.0892 | 0.0075 | 75.77 | 77.09 | 75.44 | 76.28 |
| Speckle noise | 0.0711 | 0.0686 | 0.0921 | 0.0072 | 75.55 | 76.71 | 75.36 | 76.15 |
| Impulse noise | 0.0779 | 0.0706 | 0.0988 | 0.0077 | 73.74 | 74.61 | 73.71 | 74.49 |
| Defocus blur | 0.0108 | 0.0071 | 0.0178 | 0.0256 | 92.16 | 91.55 | 92.14 | 91.63 |
| Gaussian blur | 0.0130 | 0.0116 | 0.0214 | 0.0239 | 90.79 | 89.73 | 90.67 | 90.22 |
| Motion blur | 0.0147 | 0.0103 | 0.0233 | 0.0298 | 90.33 | 90.22 | 90.20 | 90.52 |
| Zoom blur | 0.0099 | 0.0070 | 0.0185 | 0.0301 | 91.24 | 90.71 | 91.12 | 91.36 |
| snow | 0.0298 | 0.0245 | 0.0419 | 0.0208 | 86.17 | 86.11 | 86.10 | 85.81 |
| Fog | 0.0114 | 0.0075 | 0.0176 | 0.0259 | 91.67 | 91.22 | 91.64 | 91.27 |
| Brightness | 0.0081 | 0.0076 | 0.0129 | 0.0281 | 93.47 | 92.94 | 93.45 | 92.98 |
| Contrast | 0.0110 | 0.0127 | 0.0141 | 0.0306 | 91.34 | 90.59 | 91.36 | 90.73 |
| Elastic transform | 0.0244 | 0.0213 | 0.0367 | 0.0220 | 87.06 | 86.61 | 86.98 | 86.94 |
| Pixelate | 0.0350 | 0.0269 | 0.0463 | 0.0124 | 85.75 | 86.11 | 85.61 | 85.74 |
| Jpeg compression | 0.0605 | 0.0522 | 0.0813 | 0.0093 | 78.01 | 78.90 | 77.57 | 79.80 |
| Spatter | 0.0242 | 0.0163 | 0.0341 | 0.0227 | 87.94 | 87.61 | 87.99 | 88.13 |
| Saturate | 0.0112 | 0.0080 | 0.0179 | 0.0288 | 92.10 | 91.79 | 92.09 | 91.81 |
| Frost | 0.0253 | 0.0174 | 0.0365 | 0.0215 | 86.60 | 86.48 | 86.52 | 86.39 |
| **Average** | 0.0322 | **0.0283** | 0.0446 | **0.0201** | 85.65 | **85.71** | 85.52 | **85.76** |

Table 3: Log scale of Hessian eigenvalue of WideResNet28x10 training on CIFAR-100. $\lambda_1$ is the largest eigenvalue and $\lambda_5$ is 5th largest eigenvalue

| Method | $\lambda_1 \downarrow$ | $\lambda_1/\lambda_5$ |
| --- | --- | --- |
| SWAG | $4.17 \pm 0.001$ | $1.17 \pm 0.012$ |
| F-SWAG | $\mathbf{4.08 \pm 0.000}$ | $1.17 \pm 0.020$ |
| SGLD | $3.34 \pm 0.031$ | $1.17 \pm 0.009$ |
| F-SGLD | $\mathbf{2.83 \pm 0.029}$ | $1.15 \pm 0.010$ |
| Deep-ensemble | $4.64 \pm 0.055$ | $1.45 \pm 0.020$ |
| F-Deep-ensemble | $\mathbf{4.01 \pm 0.054}$ | $1.58 \pm 0.032$ |

Table 4: Comparison of training time per epoch

| Network & Dataset | SWAG | F-SWAG | SGLD | F-SGLD | Deep-ensemble | F-Deep-ensemble |
| --- | --- | --- | --- | --- | --- | --- |
| WideResNet28x10 & CIFAR-100 | 110s | 169s | 110s | 233s | 110s | 218s |
| Densenet-161 & ImageNet | 1.75h | 2.28h | 1.78h | 2.49h | - | - |
| ResNet-152 & ImageNet | 1.59h | 2.15h | 1.64h | 2.22h | - | - |

deviation. Based on the result, without KL loss, our method still manages to yield better numbers than the non-flat counterparts.

## 3 Experimental settings

**CIFAR:** We conduct experiments using PreResNet-164, WideResNet28x10, Resnet10 and Resnet18 on both CIFAR-10 and CIFAR-100. The total number of images in these datasets is 60,000, which comprises 50,000 instances for training and 10,000 for testing. For each network-dataset pair, we apply Sharpness-Aware Bayesian methodology to various settings, including F-SGLD, F-SGVB, F-SWAG-Diag, F-SWAG, F-MC-Dropout, and F-Deep-Ensemble.

In the experiments presented in Tables 1 and 2 in the main paper, we train all models for 300 epochs using SGD, with a learning rate of 0.1 and a cosine schedule. We start collecting models after epoch 161 for the F-SWA and F-SWAG settings, consistent with the protocol in [4]. Additionally, we set $\rho = 0.05$ for CIFAR-10 and $\rho = 0.1$ for CIFAR-100 in all experiments, except for Resnet10 and Resnet18, where $\rho$ is set to 0.01. The training set is augmented with basic data augmentations,

Table 5: Experiments of F-Deep-ensemble variations on CIFAR-100 dataset using WideResNet28x10. Each experiment is conducted with three different random seeds to calculate mean and standard deviation

| Model | WideResNet28x10 | | |
| --- | --- | --- | --- |
| | ACC ↑ | NLL ↓ | ECE ↓ |
| Deep-ensemble | $83.04 \pm 0.15$ | $0.6958 \pm 0.0335$ | $0.0483 \pm 0.0017$ |
| F-Deep-ensemble (**Our**) | $\mathbf{84.52 \pm 0.03}$ | $\mathbf{0.5644 \pm 0.0106}$ | $\mathbf{0.0191 \pm 0.0039}$ |
| F-Deep-ensemble (w/o L2) | $83.80 \pm 0.10$ | $0.7026 \pm 0.0007$ | $0.0594 \pm 0.0005$ |

including horizontal flip, padding by four pixels, random crop, and normalization. For the experiments presented in Table 3 in the main paper, we apply the same augmentations to the training set as in the experiments in Tables 1 and 2. However, the models are trained for 200 epochs using the Adam optimizer, with a learning rate of 0.001 and a plateau schedule. It's worth noting that SGVB and SGVB-LRT perform poorly with other settings than those mentioned, making it challenging to scale up this approach.

For the baseline of the Deep-Ensemble, SGLD, SGVB and SGVB-LRT methods, we reproduce results following the hyper-parameters and processes as our flat versions. Note that we train three independent models for the Deep-Ensemble method. For inference, we do an ensemble on 30 sample models for all settings sampled from posterior distributions. To ensure the stability of the method, we repeat each set three times with different random seeds and report the mean and standard deviation.

**ImageNet:** This is a large and challenging dataset with 1000 classes. We conduct experiments with Densenet-161 and ResNet-152 architecture on F-SWAG-Diag, F-SWAG, and F-SGLD. For all settings, we initialize the models with pre-trained weights on the ImageNet dataset, obtained from the *torchvision* package, then fine-tuned for 10 epochs with $\rho = 0.05$. We start collecting 4 models per epoch at the beginning of the fine-tuning process and evaluate them following a protocol consistent with the CIFAR dataset experiments.

The performance metrics for the SWAG-Diag, SWAG, and MC-Dropout methods are sourced from the original paper by Maddox et al. [4], except for the MC-Dropout result on PreResNet-164 for the CIFAR-100 dataset, which we reproduce due to its unavailability. The performance of bSAM is taken from [5].

It's important to note that the purpose of these experiments was not to achieve state-of-the-art performance. Instead, we aim to demonstrate the utility of the sharpness-aware posterior when integrated with specific Bayesian Neural Networks. The implementation is provided in `https://github.com/anh-ntv/flat_bnn.git`.

**Hyper-parameters for training**: Table 6 provides our setup for both training and testing phases. Note that the SWAG-Diag method follows the same setup as SWAG. Typically, using the default $\rho = 0.05$ yields a good performance across all experiments. However, $\rho = 0.1$ is recommended for the CIFAR-100 dataset in [1]. For model evaluation, we use the checkpoint from the final epoch without taking into account the validation set's performance.

Table 6: Hyperparameters for training both flat and non-flat versions of BNNs. All models are trained with the input resolution of $224 \times 224$ and cosine learning rate decay, except experiments of SGVB and SGVB-LRT, which use an input resolution of $32 \times 32$

| Model | Method | Init weight | Epoch | LR init | Weight decay | $\rho$ | # samples |
|---|---|---|---|---|---|---|---|
| | | **CIFAR-100** | | | | | |
| PreResNet-164 | SWAG | | | | | | 30 |
| | MC-Drop | Scratch | 300 | 0.1 | 3e-4 | 0.1 | 30 |
| | Deep-Ens | | | | | | 3 |
| WideResNet28x10 | SWAG | | | | | | 30 |
| | MC-Drop | Scratch | 300 | 0.1 | 5e-4 | 0.1 | 30 |
| | Deep-Ens | | | | | | 3 |
| Resnet10 | SGVB | | | | | 5e-3 | |
| | F-SGVB + Geometry | | | | | 5e-4 | |
| | SGVB-LRT | Scratch | 200 | 0.001 | 5e-4 | 5e-3 | 30 |
| | F-SGVB-LRT + Geometry | | | | | 5e-4 | |
| Resnet18 | SGVB | | | | | 5e-3 | |
| | F-SGVB + Geometry | | | | | 5e-4 | |
| | SGVB-LRT | Scratch | 200 | 0.001 | 5e-4 | 5e-3 | 30 |
| | F-SGVB-LRT + Geometry | | | | | 5e-4 | |
| | SWAG | Scratch | 300 | 0.1 | 5e-4 | 0.1 | 30 |
| | | **CIFAR-10** | | | | | |
| PreResNet-164 | SWAG | | | | | | 30 |
| | MC-Drop | Scratch | 300 | 0.1 | 3e-4 | 0.05 | 30 |
| | Deep-Ens | | | | | | 3 |
| WideResNet28x10 | SWAG | | | | | | 30 |
| | MC-Drop | Scratch | 300 | 0.1 | 5e-4 | 0.05 | 30 |
| | Deep-Ens | | | | | | 3 |
| Resnet10 | SGVB | | | | | 5e-3 | |
| | F-SGVB + Geometry | | | | | 5e-4 | |
| | SGVB-LRT | Scratch | 200 | 0.001 | 5e-4 | 5e-3 | 30 |
| | F-SGVB-LRT + Geometry | | | | | 5e-4 | |
| Resnet18 | SGVB | | | | | 5e-3 | |
| | F-SGVB + Geometry | | | | | 5e-4 | |
| | SGVB-LRT | Scratch | 200 | 0.001 | 5e-4 | 5e-3 | 30 |
| | F-SGVB-LRT + Geometry | | | | | 5e-4 | |
| | SWAG | Scratch | 300 | 0.1 | 5e-4 | 0.1 | 30 |
| | | **ImageNet** | | | | | |
| DenseNet-161 | All methods | Pre-trained | 10 | 0.001 | 1e-4 | 0.05 | 30 |
| ResNet-152 | All methods | Pre-trained | 10 | 0.001 | 1e-4 | 0.05 | 30 |