# OpenReview forum: "Flat Seeking Bayesian Neural Networks"
_NeurIPS.cc/2023/Conference — NeurIPS 2023 poster_

### Official Review · Reviewer_FrCB · 2023-06-28

**Soundness:** 3 good
**Presentation:** 2 fair
**Contribution:** 3 good
**Rating:** 6
**Confidence:** 3

**Summary:**

This paper proposes modifying the loss used for Bayesian neural networks (BNNs) to take into account the sharpness/flatness of the loss with respect to the model parameters. Theory, based on that of sharpness-aware minimization (SAM), is developed to propose this loss modification. Making BNNs sharpness-aware led to increased accuracy, decreased negative log-likelihood (NLL), and decreased expected calibration error (ECE) for large neural network models across CIFAR-10, CIFAR-100, and ImageNet.

**Strengths:**

This paper does an excellent job of combining sharpness-aware and BNN methods. The theory for this is given and the theory is applied to several BNN method, namely stochastic weight averaging Gaussian (SWAG), stochastic gradient Langevin dynamics (SGLD), Monte Carlo (MC) dropout, and deep ensembles, for CIFAR-10 and CIFAR-100. These experiments show reasonable improvements.

**Weaknesses:**

1. The difference between the optimization problems used in the different Bayesian methods was not clear enough. The main optimization problem discussed was for variational inference. To my knowledge, however, SWAG and SGLD typically use a procedure that is based on the gradient of the log of the posterior.

2. The incomplete ImageNet experiment. Only SWAG and FSWAG are evaluated for ImageNet, versus all of the methods that were applied to CIFAR-10 and CIFAR-100.

**Questions:**

1. Are different optimization problems used for some of the studied methods? If so, please add those details to Section 3.

2. Is it possible to have results on ImageNet for more of the studied methods?

3. I'd recommend putting the number of repetitions used to create the confidence intervals in the captions of the tables. I see that, in the supplementary material, it was three, but I think that this information, along with the number of samples used for the methods, should be in the main text.

4. How do you think that sharpness-aware methods would interact with Laplace approximation methods, another popular BNN method? (This is not a necessary addition to the paper, in my opinion. However, I am interested in your answer to this.)

**Limitations:**

Limitations and societal impact are not discussed. One limitation that could be added to the conclusion is that the proposed Gaussian variational approach only uses a diagonal covariance for the posterior approximation. While this is standard, it is a limitation that, by mentioning it, could show areas of potential improvement for the proposed variational method.

---

> ### Author Rebuttal · Authors · 2023-08-10
>
>
> We sincerely appreciate your constructive comments. We are dedicated to addressing all the questions listed below to the best of our capabilities.
>
> **Is it possible to have results on ImageNet for more of the studied methods?**
>
> We acknowledged the limited number of experiments on ImageNetWe and would like to report the additional results of SGLD and F-SGLD in Table 1 in the attached pdf. It's important to emphasize that the process of training models on the ImageNet dataset consumes significantly more time compared to other datasets, making it challenging to expand experiments in a limited rebuttal period. However, we wish to maximize our efforts and do as much as possible to add to the revised version.
>
> **I’d recommend putting the number of repetitions used to create the confidence intervals in the captions of the tables**
>
> We will carefully revise the paper incorporating your suggestions to enhance its quality and comprehensiveness.
>
> **How do you think that sharpness-aware methods would interact with Laplace approximation methods, another popular BNN method?**
>
> Thanks for this interesting but challenging question. A possible solution is to start from our proposed sharpness-aware posterior $p^{SA}$ and then perform second-order Taylor expansion for $\text{ln } p^{SA}(\theta \mid \mathcal{D})$. The $\theta_{MAP}$ is the SAM solution and we can resort to some techniques mentioned in the paper [1] to approximate the Hessian matrix at $\theta_{MAP}$.
>
> We can even go further with more complicated formulations by taking account of the derivative of $s(\theta) = \text{argmax}_{\theta' \in B\_{\rho(\theta)}} \mathcal{L}\_{S(\theta')}$ in the derivations, which potentially leads to a Hessian matrix.
>
> [1] Daxberger, E. et al. : Laplace redux-effortless bayesian deep learning, NeurIPS21.

---

> > ### Comment · Reviewer_FrCB · 2023-08-15
> >
> > Thank you for the additional ImageNet results.
> >
> > I'd mention SWAG and SGLD in Section 3.1 to add a link in the text between your proposed method and those methods.
> >
> > Thank you for adding the number of samples used to compute the confidence intervals to the table.
> >
> > Thank you for the information on the interaction between your sharpness-aware method and the Laplace approximation.

---

> > > ### Author Response · Authors · 2023-08-16
> > > **Mention SWAG and SGLD in Section 3.1 to add a link in the text between your proposed method and those methods**
> > >
> > > Thank you for your suggestion. We believe that mentioning it would enhance the clarity of our paper's setting, making it more accessible for readers to follow.

---

### Official Review · Reviewer_hAuQ · 2023-07-05

**Soundness:** 3 good
**Presentation:** 3 good
**Contribution:** 2 fair
**Rating:** 6
**Confidence:** 4

**Summary:**

This paper introduces a method called Sharpness-Aware Bayesian Neural Networks (SABNN) that aims to improve generalization performance on test datasets. The key idea is to replace the negative empirical loss function with the negative SAM (Sharpness-Aware Minimization) [1] loss function, enabling consideration of the flatness of the loss surface. The authors provide both theoretical analysis and empirical evidence to demonstrate that SABNN achieves better generalization compared to existing methods when evaluated on test datasets.

References

[1] Foret, P., A. Kleiner, H. Mobahi, et al. Sharpness-aware minimization for efficiently improving generalization. In International Conference on Learning Representations. 2021.

**Strengths:**

Originality
- The paper presents a theoretical demonstration of the robustness of sharpness-aware posterior inference for the entire distribution of the true dataset.
- The authors propose a straightforward variational approach that utilizes a Gaussian approximate posterior for efficient and straightforward inference.

Clarity
- The paper is well-written and provides clear explanations, making it easy to understand and follow.

**Weaknesses:**

Method
- The method described in the paper involves a straightforward and nearly direct modification of the likelihood function by replacing the negative empirical loss with the negative SAM loss function.

Experiment
- In section 4.3 of the paper, the authors mention that the sharpness scores are provided in the supplementary material. However, I was unable to locate the results in the supplementary material.
- The paper does not discuss any potential additional computational costs associated with the proposed method in comparison to the baselines.


**Questions:**

Experiment
- It is requested to provide the sharpness scores and the largest Hessian eigenvalues for the entire dataset, as well as for the SWAG and F-SWAG models. I think the eigenvalues can be calculated for the mean $\mu$ of each method.
- There is a desire for a discussion regarding potential additional computational costs associated with the proposed method in comparison to the baselines.
- The results of the Deep Ensemble, which is an ensemble of independently trained models without KL in Bayesian Deep Ensemble and with L2 regularization loss, trained with SAM loss, are also requested. The aim is to assess whether the proposed SABNN method is practically beneficial or not.

I am willing to increase my score after considering the experiment results provided in the rebuttal.

**Limitations:**

Limitation
- This paper did not address the limitation of the proposed method.
- One possible limitation can be additional computational costs.

---

> ### Author Rebuttal · Authors · 2023-08-10
>
> Thanks for your constructive comments. Based on your suggestion, we conduct some more experiments (report in the attached pdf file) and hope that we can address some of your points as presented below.
>
> **Largest Hessian eigenvalue**
> We report the log scale of the largest eigenvalue of the Hessian matrix over several methods applying to WideResNet28x10 on CIFAR100, and the ratio of the largest and fifth eigenvalue as shown in Table 4 in the pdf attached, which evidently indicates that our method updates models to minima having lower curvature.
>
> **The sharpness scores**
> We report the sharpness scores of the PreResNet-164 network for SWAG, and F-SWAG training on CIFAR100 in Figure 1 of the attached pdf. This shows that our proposed approach gains better sharpness scores than the baseline.
>
> **Computational cost**
>
> We further report the computational cost of several experiments in Tables 1 and 4 in the attached pdf. Similar to SAM, our flat-seeking method involves the computation of gradients twice. The initial computation is to derive the perturbed model $\theta'$ and the subsequent one is to update the model. As a result, the training time is nearly double in comparison to non-flat counterparts. We will explicitly discuss this in the limitation section.
>
> **Additional experiments on without the regularization term for the deep ensemble**
> We present the result of training Deep-ensemble with SAM following the formula for the flat version (Section 2 supplementary) _without KL (or L2 regularisation)_ in Table 2 of the attached pdf. Each experiment is performed three times and the mean and standard deviation are reported. Based on the result, the flat version *with KL divergence* performs better than the one *without KL divergence*.

---

> > ### Author Response · Authors · 2023-08-15
> > **Please kindly let us know if our additional experiments address your concerns**
> >
> > Dear Reviewer,
> >
> > Could you please look at our rebuttal and kindly let us know if the additional experiments address your concerns? Really appreciate your time and effort to review our paper.
> >
> > Regards,
> >
> > Authors of the paper 6164

---

> > > ### Comment · Reviewer_hAuQ · 2023-08-16
> > >
> > > Thank you for clarifying and sharing the supplementary experiments. I still have a few lingering inquiries regarding the costs associated with additional training. Could you present the performance results of the baseline methods under the same additional training expenses? I believe this data would serve as empirical proof that the performance improvement observed in the proposed method is not attributed to the extended training epochs.

---

> > > > ### Author Response · Authors · 2023-08-17
> > > > **Training longer with SGD optimizer**
> > > >
> > > > Thanks for your suggestion for a fairer comparison. Similar to SAM, due to the double gradient computation, our proposed approaches tend to nearly double the training times comparing to the corresponding baselines. As you suggested, we double the number of epochs for the baselines and report the predictive performance and training time. As seen in the following table, increasing the number of epochs helps slightly improve the accuracies. However, it’s notable that the NLL and ECE tend to be higher. In all cases, the overall scores are still lower than our Flat-BNN, showing the effectiveness of our proposed approaches.
> > > >
> > > >
> > > >
> > > > _Table 5. Comparison of experiments training with different numbers of epochs on CIFAR100 with PreResNet164. All experiments are conducted three times and report the mean and standard deviation, except SGLD-300e and SWAG-600e, we only conduct one time and report 0.00 as the standard deviation_
> > > > |Method         | Acc       |   NLL    |   ECE     |
> > > > |---------------|:---------:|:---------:|:----------:|
> > > > |SGLD (150 epochs)      | 	80.13 $\pm$  0.01 |	0.7604 $\pm$ 0.0010 |   0.1161 $\pm$  0.0031|
> > > > |SGLD (300 epochs)      |	80.30  $\pm$ 0.00 |	0.7549  $\pm$ 0.0000  |   0.1181   $\pm$ 0.0000 |
> > > > |F-SGLD (150 epochs)    |	**80.82 $\pm$ 0.02**   |	**0.7276 $\pm$ 0.0012**  |   **0.1085 $\pm$ 0.0008** |
> > > >  |----------- |---- |----- | ----- |
> > > > |SWAG (300 epochs)       |	79.9 $\pm$ 0.50    |	**0.6595 $\pm$ 0.0019**  |   0.0587 $\pm$ 0.0048  |
> > > > |SWAG (600 epochs)       |	80.52  $\pm$ 0.00   |	0.8736   $\pm$ 0.0000 |	0.1992  $\pm$ 0.0000 |
> > > > |F-SWAG (300  epochs)     |	**80.93 $\pm$ 0.27**   |	0.6704 $\pm$ 0.0049  |	**0.0350 $\pm$ 0.0025**   |

---

> > > > > ### Comment · Reviewer_hAuQ · 2023-08-17
> > > > >
> > > > > Your inclusion of the supplementary experiments is appreciated. Please incorporate all the extra experimental findings into the revised paper. Considering the supplementary experiments and their results, I am inclined to raise my evaluation score for acceptance.

---

> > > > > > ### Author Response · Authors · 2023-08-17
> > > > > >
> > > > > > Thank you for your consideration and helpful responses. We really appreciate it, and we will certainly include the additional experiments and analysis in our revised version to enhance the strength of our paper.

---

### Official Review · Reviewer_yGhH · 2023-07-06

**Soundness:** 3 good
**Presentation:** 1 poor
**Contribution:** 3 good
**Rating:** 5
**Confidence:** 2

**Summary:**

This paper extends SAM -- sharpness aware minimization framework of Foret et al, who seeks parameters of Neural Networks in the regimes of flat loss landscape. The current understanding is that a model that exhibits a flat loss landscape exhibits better generalization performance. Particularly in this paper, such methods are extended to variational inference for neural networks, where the posterior (and hence neural network training) is obtained through an objective function that seeks flat local minima. Experiments are conducted, which were designed to incorporate their objective function into the current Bayesian Neural Networks. The results show consistent improvements over the vanilla counterparts.

**Strengths:**

The contribution of the paper seems interesting. The framework is utilizing PAC-Bayes like theory, where an upper bound to the true loss function is obtained through a careful balance between empirical loss and certain KL divergence. The later regularizes the Bayesian neural network overfitting. It is interesting to see, how variational inference fits the true posterior, which is in turn modified according to the flat local minima seeking objectives. As SAM is one of the popular methods to understand generalization (while being simple), such extensions to Bayesian Neural Networks might be meaningful to the community.

**Weaknesses:**

I am not an expert in this specific area of combining geometric insights into variational inference. I lean to accept the paper due to my inability to fully grasp this concept. I hope that other reviewers can provide more meaningful feedback for improvements here.

For the weakness:

- presentation could perhaps improve a bit more.

1. in the abstract, I do not see the logical connection between generic explanations of Bayesian Neural Networks, and generalization. The switch was to me, a bit disconnected and sounded a bit sudden. In the introduction, I also did not see much the role of the 2nd paragraph. Indeed, there have been many variation inference approaches, but I could not see the connection between the expressivity of the posterior, and the approaches of seeking flatness in variational inference.

2. in the theoretic part, I see several equations that repeat also in other papers, e.g., the upper bound of true loss with empirical and KL divergence, how the true posterior provides tight generalization bound, etc. While I appreciate the author's efforts, especially in the appendix for many derivations, it may also make sense to provide citations more in the corresponding texts, so that the readers may be able to learn also through other papers.

3. lines 170-180 should be extended a bit more. I could not understand the connection between each question on page 5 and also missed insights between each derivation.

4. It might be helpful to include an algorithm part, so that the readers may better comprehend how the method works (and hopefully also why).

- empirical improvements seem rather minor, and it was not clear to me, how these experiments demonstrate generalization.

While I appreciate the ample empirical evidence provided by the paper, most of the improvements in terms of accuracy are less than 1 percent. This may raise a question if the experimental design here is adequate to show the generalization performance. Table 3, 4, and 5 need error bars, which is requirements in Bayesian machine learning works.

On how to test generalization, I think the paper could adapt few-shot settings or experiments from meta-learning. Therein, little training data is provided, and one may be able to test better how the model still generalizes. I am not sure, how testing on vanilla cifar10, cifar100 or imagenet classification score is an indication of generalization performance.

**Questions:**

--

**Limitations:**

I could not find the limitation section in the main body of the paper.

---

> ### Author Rebuttal · Authors · 2023-08-10
>
> We sincerely appreciate your constructive comments. We carefully address all the questions listed below to the best of our capabilities and improve our paper based on that.
>
> **Presentation could perhaps improve a bit more.**
>
> Thank you for pointing this out. We will enhance the clarity and motivation behind the fusion of Bayesian Neural Networks and the concept of generalization in the revised version.
>
> **Provide citations more in the corresponding texts, so that the readers may be able to learn also through other papers.**
>
> Thank you for recognizing our efforts in the theoretical derivations. We will definitely provide citations for equations that have also been explored in other works. This will offer readers a broader context for understanding our extensions.
>
> **I could not understand the connection between each question on page 5 and also missed insights between each derivation.**
>
> We apologize for any confusion caused. We will begin by presenting the proof of Theorem 3.2, which extends the concept of sharpness-aware minimization to general and empirical losses over a distribution Q.
>
> In the following derivation, we modify the upper-bound of the inequality to explicitly represent _the sharpness over the distribution Q in the model space_. This modification aims to offer readers a clearer understanding of the concept of sharpness-aware minimization over Q.
> Notably, the upper-bound presented in Equation (5) is directly influenced by the insights from Theorem 3.2.
> In the revised version, we will ensure that these connections are articulated more comprehensively to enhance clarity.
>
> **Include an algorithm part**
>
> We appreciate your suggestion. While we view our approach as a general framework applicable to various algorithms and settings, we recognize the importance of including specific algorithms for better comprehension. In the revised version, we will present algorithms for different settings to enhance clarity.
>
>
> **Empirical improvements seem rather minor, and it was not clear to me, how these experiments demonstrate generalization. Table 3, 4, and 5 need error bars, which is requirements in Bayesian machine learning works.**
>
> We understand the importance of the error bar for each experiment in the BNN field and try our best to provide as much as possible in the rebuttal time. Due to the limited rebuttal time, we can only report the error bar for Gaussian posterior experiments as shown in Table 3 in the attached pdf. We will try our best to report the error bars for our approaches and the baselines in the revised version. Meanwhile, we are able to conduct one more setting of SGLD and F-SGLD on ImageNet (Table 1 in attached pdf) and also report the largest eigenvalue of the Hessian matrix over several methods, and the ratio of the largest and fifth eigenvalue (Table 4 in attached pdf) as additional evidence that our method updates models to minima having lower curvature (based on the connection between the geometry of the loss landscape
> and generalization, which has been studied extensively [1]).
>
> [1] Gintare Karolina Dziugaite and Daniel M Roy. Computing non vacuous generalization bounds for deep (stochastic) neural networks with many more parameters than training data. arXiv preprint arXiv:1703.11008, 2017.
>
> **Adapt few-shot settings or experiments from meta-learning.**
>
> Thank you for your suggestion on the evaluation of generation under alternate settings. They are directly related to generalization ability, hence potentially applying our approaches, however, we leave it for future works. Currently, our paper mainly focuses on experiments on the Bayesian Neural Network and the effectiveness of sharpness aware posterior. The effectiveness is demonstrated through the consistence improvement of accuracy, which is a reasonable measurement of generalization ability. In Table 5 in the main paper, we also evaluate our proposed approach against the non-flat versions in an out-of-distribution setting. Particularly, the original images added different types of corruption and noise. The consistent improvement in both overall accuracy and ECE is additional evidence of the generalization ability of our proposed methods.

---

> > ### Comment · Reviewer_yGhH · 2023-08-16
> > **Response to the authors**
> >
> > I would like to thank the authors for the efforts. I have read the rebuttal and comments from other reviewers.
> >
> > I decided to stand with the score of borderline. More thoughtful experiments, which better showcase the strength of the methods could make the paper more relevant to the community.

---

> > > ### Author Response · Authors · 2023-08-21
> > >
> > > We respect your decision. Thank you for taking the time to review our paper and hope we have a chance to improve it in the revised version based on your suggestion.

---

### Official Review · Reviewer_VfFA · 2023-07-08

**Soundness:** 2 fair
**Presentation:** 2 fair
**Contribution:** 2 fair
**Rating:** 5
**Confidence:** 3

**Summary:**

The paper proposes a new approach to posterior inference for Bayesian neural networks that takes into account the sharpness/flatness of deep learning models, leading to better generalisation ability. The authors introduce the Sharpness-Aware Posterior (SA-Posterior), which allows the sampling of a set of flat models that improve model generalisation. The paper presents a theoretical framework for the SA-Posterior, including a Bayesian framework and a variational inference approach. The authors demonstrate the effectiveness of their approach through experiments on various datasets, showing that it outperforms existing methods on all metrics of interest.

**Strengths:**

- The paper presents a novel approach to posterior inference for Bayesian neural networks.
- The theoretical framework for the SA-Posterior is developed, including a Bayesian setting and a variational inference approach.
- The effectiveness of the SA-Posterior is demonstrated through experiments on various datasets.
- The paper provides insights into the importance of sharpness/flatness in deep learning models.
- The paper contributes to the growing body of research on improving the generalisability of deep learning models.

**Weaknesses:**

- The experiments could be extended to include more datasets and models.
- The paper does not discuss the computational complexity of the SA-Posterior approach.
- The paper does not compare the SA-Posterior approach with other recent approaches to improve the generalisation ability of deep learning models.

**Questions:**

- How does the SA-Posterior approach compare to other recent approaches to improve generalisation in deep learning models?
- How does the computational complexity of the SA-Posterior approach compare with other approaches to posterior inference for Bayesian neural networks?

**Limitations:**

- Investigate the potential of the SA posterior approach for other types of neural networks beyond Bayesian neural networks.
- Investigate the computational complexity of the SA-Posterior approach and develop methods to reduce it.
- Investigate the potential of the SA-Posterior approach for transfer learning and domain adaptation.

---

> ### Author Rebuttal · Authors · 2023-08-10
>
> We sincerely appreciate your constructive comments. We carefully address all the questions listed below to the best of our capabilities and improve our paper based on that.
>
> **The experiments could be extended to include more datasets and models.**
>
> Thanks for this comment. In this paper, we conducted the experiments on three datasets including Cifar10, Cifar100, and ImageNet, and compared with 9 baselines including MC-Dropout, Deep-ens, SGLD, SWAG, SWAG-Diag, SGVB, SGVB-LRT, SAM, and bSAM. We believe this current amount of experiments satisfies the standard of our field.
>
> Encouraged by your suggestion, we conduct more experiments on ImageNet for more baselines as reported in Table 1 in the attached pdf. Additionally, we report the largest eigenvalue of the Hessian matrix over several methods applying to WideResNet28x10 on CIFAR100, and the ratio of the largest and fifth eigenvalue (Table 4 in the attached pdf) as evidence that our method updates models to minima having lower curvature.
>
> **The paper does not discuss the computational complexity of the SA-Posterior approach**
>
> We further report the computational cost of several experiments in Tables 1 and 4 in the attached pdf. Similar to SAM, our flat-seeking method involves the computation of gradients twice. The initial computation is to derive the perturbed model $\theta'$ and the subsequent one is to update the model. As a result, the training time is nearly double in comparison to non-flat counterparts. We will explicitly discuss this in the limitation section.
>
> **The paper does not compare the SA-Posterior approach with other recent approaches to improve the generalisation ability of deep learning models**
>
> We provide the comparison of our proposed methods with _bSAM_ and _SAM_ (cf. Table 6 in supplementary), which are recent works using sharpness-aware minimization for improving the generalization ability of deep nets. The results demonstrate that our flat version of BNN outperforms bSAM in the most of metric scores.

---

> > ### Comment · Reviewer_VfFA · 2023-08-19
> >
> > I would like to thank the authors for their detailed response.
> > My concerns have been addressed and I will keep my score.

---

> > > ### Author Response · Authors · 2023-08-21
> > >
> > > We respect your decision. Thank you for taking the time to review our paper and hope we have a chance to improve it in the revised version based on your suggestion.

---

### Official Review · Reviewer_P465 · 2023-07-12

**Soundness:** 3 good
**Presentation:** 2 fair
**Contribution:** 3 good
**Rating:** 6
**Confidence:** 2

**Summary:**

This paper introduce theories in Bayesian settings and  propose variational inference for the sharpness aware posterior in the context of Bayesian Neural Network. The proposed approach is incorporated with existing state of the art Bayesian Neural Networks and experiments were conducted to show the effectiveness of sharpness aware posterior. The results indicate that incorporating sharpness aware posterior methodolody outperforms baseline in terms of ensemble accuracy , expected callibration error and negative likelihood. Also, experimensts show that models from poroposed approach are less sensitive to noise  and have improved generalization ability

**Strengths:**

1) The paper proposes to approximate posterior of Bayesian Neural Networks with sharpness aware posterior.
2) The proposed approach can easily be incorporated to already existing Bayesian Neural Networks
3) The experiments use state-of-the-art settings
4) The paper outperforms the baseline methods

**Weaknesses:**

1) The experments show that proposed approach most of teh times outperforms baseline methods , however, improvement is not that significant.
2) The experimenst are only for image-classification task



**Questions:**

Q1) What is the computational head of the proposed approach?

**Limitations:**

It would be great to add compared to baseline approaches , what is the computational overhead of the proposed approach.

---

> ### Author Rebuttal · Authors · 2023-08-10
>
>
> We sincerely appreciate your constructive comments. We carefully address your question to the best of our capabilities and improve our paper based on that.
>
> **Computational head**
>
> We further report the computational cost of several experiments in Tables 1 and 4 in the attached pdf. Similar to SAM, our flat-seeking method involves the computation of gradients twice. The initial computation is to derive the perturbed model $\theta'$ and the subsequent one is to update the model. As a result, the training time is nearly double in comparison to non-flat counterparts. We will explicitly discuss this in the limitation section.

---

> > ### Comment · Reviewer_P465 · 2023-08-19
> >
> > I would like to thank the authors for the many clarifications.

---

> > > ### Author Response · Authors · 2023-08-21
> > >
> > > Thank you again for taking the time to review our paper and hope our responses effectively address your concerns.

---

### Author Rebuttal · Authors · 2023-08-10

We appreciate the reviewers' constructive comments. We would like to report additional experiments on both ImageNet and CIFAR datasets, the computation cost, the sharpness scores, and the eigenvalues of the Hessian matrix in the attached pdf.

---

### Decision · Program_Chairs · 2023-09-21

**Decision:**

Accept (poster)

**Comment:**

The paper introduces an approach to posterior inference for Bayesian Neural Networks (BNNs) that considers the sharpness/flatness of deep learning models. The primary contribution is the Sharpness-Aware Posterior (SA-Posterior), which targets flat models, aiming to boost model generalization. The proposed method is theoretically grounded and demonstrates improvements over existing techniques via experiments conducted on datasets like CIFAR-10, CIFAR-100, and ImageNet.

**Strengths of the paper:**
- Proposes a novel combination of sharpness-aware and Bayesian neural network methodologies.
- The theoretical foundation is robust, relying on the principles of sharpness-aware minimization.
- Can be easily integrated into existing Bayesian Neural Network setups.
- Demonstrates empirically that it surpasses baseline methodologies across several datasets.

**Weaknesses of the paper:**
- The improvement over baseline methods, although consistent, is incremental or not statistically significant in many cases.
- Experimental setup is primarily limited to image classification tasks and is not shown to scale up to ImageNet. Fortunately, this is partially mitigated by new results in the rebuttal.
- Computational complexity and overhead when compared to baseline methods aren't addressed.
- Some reviewers highlighted issues in presentation, stating that parts of the paper were not clear enough or seemed disconnected.

The paper makes a meaningful contribution to the field and provides valuable insights. It could benefit from more thorough experiments and clearer presentation of some concepts. I recommend the authors incorporate their new results from the rebuttal into a revision and polish the writing in order to improve clarity.